# Assessing the ecological patterns of *Aedes aegypti* in areas with high arboviral risks in the large city of Abidjan, Côte d'Ivoire

Claver N. Adjobi[1,2]*, Julien Z. B. Zahouli[2,3]*, Négnorogo Guindo-Coulibaly[1], Allassane F. Ouattara[2,4], Laura Vavassori[5,6], Maurice A. Adja[1]

**1** Laboratoire de Biologie et Santé, UFR Biosciences, Université Félix Houphouët-Boigny, Abidjan, Côte d'Ivoire, **2** Centre Suisse de Recherches Scientifiques en Côte d'Ivoire, Abidjan, Côte d'Ivoire, **3** Centre d'Entomologie Médicale et Vétérinaire, Université Alassane Ouattara, Bouaké, Côte d'Ivoire, **4** Unité de Formation et de Recherche Sciences de la Nature, Université Nangui-Abrogoua, Abidjan, Côte d'Ivoire, **5** Swiss Tropical and Public Health Institute, Allschwill, Switzerland, **6** University of Basel, Basel, Switzerland

* claver.adjobi@csrs.ci (CNA); julien.zahouli@csrs.ci (JZBZ)

## Abstract

### Background

The city of Abidjan, Côte d'Ivoire has increasingly faced multiple outbreaks of *Aedes* mosquito-borne arboviral diseases (e.g., dengue (DEN) and yellow fever (YF)) during the recent years, 2017–2024. Thus, we assessed and compared *Aedes aegypti* larval and adult population dynamics and *Stegomyia* indices in four urbanized areas with differential arboviral incidences in Abidjan, Côte d'Ivoire.

### Methods

From August 2019 to July 2020, we sampled *Aedes* mosquito immatures (larvae and pupae), adults and breeding habitats in Anono and Gbagba with high arboviral incidences and Ayakro and Entente with low arboviral incidences in the Abidjan city, using standardized methods. Sampling was conducted in the peridomestic and domestic (indoors and outdoors) premises during short dry season (SDS), short rainy season (SRS), long dry season (LDS) and long rainy season (LRS). The abdomens and ovaries of *Ae. aegypti* females were examined to determine their blood-meal and parity statuses. *Stegomyia* indices (container index: CI, house index: HI and Breteau index: BI), blood-meal status and parity rates were compared by study sites and seasons and with the World Health Organization (WHO)-established epidemic thresholds.

### Results

Overall, *Aedes* and arboviral risk indices were high and similar between the four study areas. In total, 86,796 mosquitoes were identified and dominated by *Ae. aegypti* species (97.14%, 84,317/86,796). The most productive larval breeding habitats were tires, discarded containers and water storage containers. CI, HI, and BI in Anono (22.4%, 33.5% and 89.5), Ayakro (23.1%, 43.8% and 91.0), Entente (15.9%, 24.8% and 48.5) and Gbagba (23.3%, 43.0% and 102.0) were high in the respective study sites. *Stegomyia* indices were

**Data Availability Statement:** All relevant data are within the paper and its Supporting Information files.

**Funding:** The author(s) received no specific funding for this work.

**Competing interests:** The authors have declared that no competing interests exist.

higher than the WHO-established epidemic thresholds during any seasons for DEN, and LRS and SRS for YF. The numbers of *Ae. aegypti*-positive breeding sites were higher in the domestic premises (68.0%, 900/1,324) than in the peridomestic premises (32.0%, 424/1,324). In the domestic premises, *Ae. aegypti*-positive breeding sites (94.6%, 851/4,360) and adult individuals (93.4%, 856/916) were mostly found outdoors of houses. *Aedes aegypti* adult females were mostly unfed (51.3%, 203/396), followed by blood-fed (22.2%, 88/396), gravid (13.9%, 55/396) and half-gravid (12.6%, 50/396), and had parity rate of 49.7% (197/396) that was comparable between the study sites.

## Conclusions

The city of Abidjan, Côte d'Ivoire is highly infested with *Ae. aegypti* which showed comparable ecological patterns across study sites and seasons. Thus, the local communities are exposed to high and permanent risks of transmission of DEN and YF viruses that were above the WHO-established epidemic thresholds throughout. The results provide a baseline for future vector studies needed to further characterize the observed patterns of local *Ae. aegypti* abundances and behaviors, and risks of transmission of these arboviruses. Community-based larval source management of identified productive containers might reduce *Ae. aegypti* numbers and risks of transmission of *Aedes*-borne arboviruses in Abidjan, and other sub-Saharan African cities.

### Author summary

As most sub-Saharan African cities, Abidjan in Côte d'Ivoire has faced recently a considerable increase in the outbreaks of dengue (DEN) and yellow fever (YF). However, critical data are still lacking on the ecology of the main vector *Aedes aegypti* and the risk of transmission of DEN and YF viruses. We assessed the ecology of *Ae. aegypti* mosquitoes and the risk of DEN and YF virus transmission in areas with high and low DEN and YF occurrences in Abidjan. Our findings revealed a significant presence of *Ae. aegypti*, indicating a high risk of YF and DEN transmission across all study areas. *Ae. aegypti* larvae were mostly breeding in tires, discarded containers and water storage containers. Meanwhile, elderly adult females were abundant, alongside a significant presence of both unfed host-seeking and blood-fed individuals in and around residences. We observed seasonal risk patterns, with permanent and high threats for DEN observed throughout both rainy and dry seasons over the entire year, alongside elevated risks of YF during both short and long rainy seasons across all study areas. Our findings provide insights into the ecology of *Ae. aegypti* and the epidemiology of DEN and YF, crucial for strategically targeting and controlling this vector in areas both affected and unaffected by the disease outbreaks. Community-based intervention programs for managing identified larval breeding sites might reduce *Ae. aegypti* numbers to prevent future outbreaks of DEN and YF in Abidjan and other sub-Saharan African cities.

## Introduction

*Aedes* mosquito-borne arboviral diseases such as dengue (DEN), yellow fever (YF), chikungunya (CHIK) and Zika (ZIK), pose a significant threat to over 831 million people, representing

70% of population in sub-Saharan Africa [1]. These diseases have the heaviest public health and socio-economic impacts in urbanized cities [2,3]. Moreover, there is an ongoing resurgence and geographical expansion of arboviral diseases [1,4], intensified by a rapid urbanization, climate change and international mobility and trade [5,6]. West Africa, including Côte d'Ivoire, is one of most important emerging and re-emerging foci and hotspot of arboviruses in Africa [7,8]. In the West African region, over 27,000 arboviral cases were reported between 2007 and 2020, with the highest incidences and greatest burdens observed in major capital cities [7]. In 2023, 171,991 suspected cases of DEN, including 70,223 confirmed and probable cases and 753 deaths have been reported from 15 African countries [9]. A neighboring country of Côte d'Ivoire, Burkina Faso, stands out as the most impacted country with 146,878 suspected cases and 688 deaths [9]. In the absence of licensed vaccines for most arboviruses (except for the YF vaccine) and the lack of widespread routine prophylactic programs for controlling *Ae.* vectors, surveillance of the primary vector, *Ae. aegypti* is crucial for preventing, controlling, responding to, and preparing for to prevent arboviral outbreaks. *Ae. aegypti*, a key vector of arboviruses in Africa [1], can transmit over 5 viruses to humans [10] and exhibits highly anthropophagic behavior, dwelling in and around human habitats such as domestic and peridomestic premises where females predominantly feed on humans and breed in man-made container [9]. The ecological adaptability of *Ae. aegypti* allows this species to colonize various breeding sites in close proximity to human dwellings [11].

Since 1898 up to the present year (2024), Côte d'Ivoire has experienced multiple outbreaks of YF and DEN, with a notable resurgence and increase incidence in recent years, particularly 2017 to the present [12–18]. Despite the historical and present backgrounds, arboviruses remain uncontrolled, with ongoing resurgence of outbreaks posing a significant public health concern, particularly evident in the densely populated and highly urbanized city of Abidjan, Côte d'Ivoire. The urban environments of Abidjan are permissive to *Ae. aegypti*. The city is marked by rapid, uncontrolled urbanization and complex land cover changes, driven by poor urban planning and limited environmental and sanitation management services. As a result, Abidjan harbors large numbers of *Ae. aegypti* (~100% of *Aedes* mosquitoes) and larval habitats, and has often faced multiple outbreaks of arboviruses (e.g., DEN and YF) [13]. There are currently no specific programs for the routine controls of arboviruses and their vectors. This is largely due to restricted of financial investments, along with limited operational resources and technical capacities. Moreover, *Ae. aegypti* in Abidjan are resistant to most insecticides used for their control [14]. The government's response to outbreaks, led by the National Institute of Public Hygiene (NIPH) under the Ministry of Health and Public Hygiene (MHPH) of Côte d'Ivoire, primarily relies on sporadic insecticide space spraying targeting adults *Aedes* mosquitoes, along with and systematic removal, physical destruction and/or treatment of larval breeding sites. The interventions, largely unplanned due to the absence of robust data and accurate predictions, are urgently implemented in response to sudden arboviral outbreaks. However, they frequently yield limited and short-term impacts on local *Aedes* vector and arboviral control efforts [13–18]. Indeed, the local *Ae. aegypti* populations recover quickly and arboviruses re-emerge in the intervention areas once the dedicated campaigns are over, as observed in 2017, 2019, 2022, 2023 and 2024 [13–18]. In 2017, Abidjan has recorded outbreaks of DEN (623 suspected, 192 confirmed and 2 fatal cases) [13]. Out of the 192 confirmed DEN cases, 66% were virus serotype 2 (DENV-2), 29% were DENV-3 and 5% were DENV-1. In 2019, outbreaks of DEN (3,201 suspected, 281 confirmed and 2 fatal cases) and YF (89 confirmed and 1 fatal cases) were reported [15,16]. In 2022, Abidjan has faced an outbreak of DEN (181 suspected, 19 confirmed and 1 fatal cases) [16] and in 2023 to an outbreak of DEN that has caused 73 infected cases and 2 deaths [18]. However, reports indicated that arboviral occurrences have shown geographical and seasonal disparities, with the majority (80–90% cases) of cases

being recorded in the health districts of Cocody and Bingerville and during rainy seasons (April-July and September-October), while very few numbers of cases being reported in the other eight health districts of Abidjan city, including Treichville and Yopougon. *Ae. aegypti* populations are highly prevalent (100% of *Aedes* genus) in Abidjan, and larvae mostly breed in discarded items (e.g., cans, tires) and water storage containers [19–21]. The current study aimed at assessing the ecology of the *Ae. aegypti* vector in four different sites with different DEN and YF incidences within the city of Abidjan: Anono in Cocody and Gbagba in Bingerville with high DEN and YF incidence (80–90% cases), and Ayakro in Yopougon and Entente in Treichville with low DEN and YF incidence (<10% cases). We hypothesized that communities are exposed to higher entomological risks of transmission of DEN and YF viruses in Anono and Gbagba (areas with high DEN and YF incidences) compared with Ayakro and Entente (areas with low DEN and YF incidences). We monitored *Ae. aegypti* populations at different development stages (i.e., larvae, pupae and adults) and larval breeding sites in the field using sensitive and standardized methods to test this hypothesis.

## Methods

### Ethics statement

Before starting the study, the study protocol received ethical approval from the National Ethical Committee (Comité National d'Ethique des Sciences de la Vie et de la Santé) Ministry of Health and Public Hygiene, Côte d'Ivoire (ref: 034-21/MSHP/CNESV5-km). Additionally, authorizations were obtained from the local administrative and health authorities. The local community leaders provided oral informed consent as well. Mosquito collections in households were done with the permission and written informed consent of the owners and/or residents. This study did not involve endangered or protected species.

### Study area

The study was conducted in the city of Abidjan (05˚ 19' N and 4˚ 01' W) located in southern Côte d'Ivoire (West Africa) (Fig 1). Abidjan is the first and the third largest city of Côte d'Ivoire and West Africa, respectively [22]. The population is estimated at 7 million inhabitants [23]. Abidjan has ten administrative municipalities, including Bingerville, Cocody, Treichville and Yopougon.

  *Ae. aegypti* species and larval breeding sites are highly abundant and ubiquitous in Abidjan [24]. Abidjan has faced multiple outbreaks of DEN and YF. However, the arboviral occurrences and incidences significantly differed from one municipality to another [13–18]. In this study, four urban municipalities were selected based on their epidemiological backgrounds and incidences of DEN and YF: Bingerville (5˚ 21' N; 3˚54' W) and Cocody (5˚ 20' N; 3˚ 58' W), located in the Cocody-Bingerville health district where DEN and YF cases are regularly reported; Treichville (05˚ 19' N; 04˚ 01' W), situated in the Marcory-Treichville district with few recorded cases of DEN and YF; and Yopougon (5˚ 20' N; 4˚ 00' W), where DEN and YF cases are rare. Cocody-Bingerville health district is well known as the main focus arboviral outbreaks of Côte d'Ivoire. Cocody-Bingerville health district accounted for over 80–90% of cases DEN and YF reported between 2017 to 2024, according to hospital data.

  Abidjan has a humid and sub-equatorial climate, characterized by four seasons: two rainy seasons from April to July (long rainy season: LRS) and from October to November (short rainy season: SRS) and two dry seasons from December to March (long dry season: LDS) and from August to September (short dry season: SDS). The annual average temperature is around 26–28˚C and the annual relative humidity ranges between 75 and 90%. The average annual precipitation ranges between 1000 and 1200 mm.

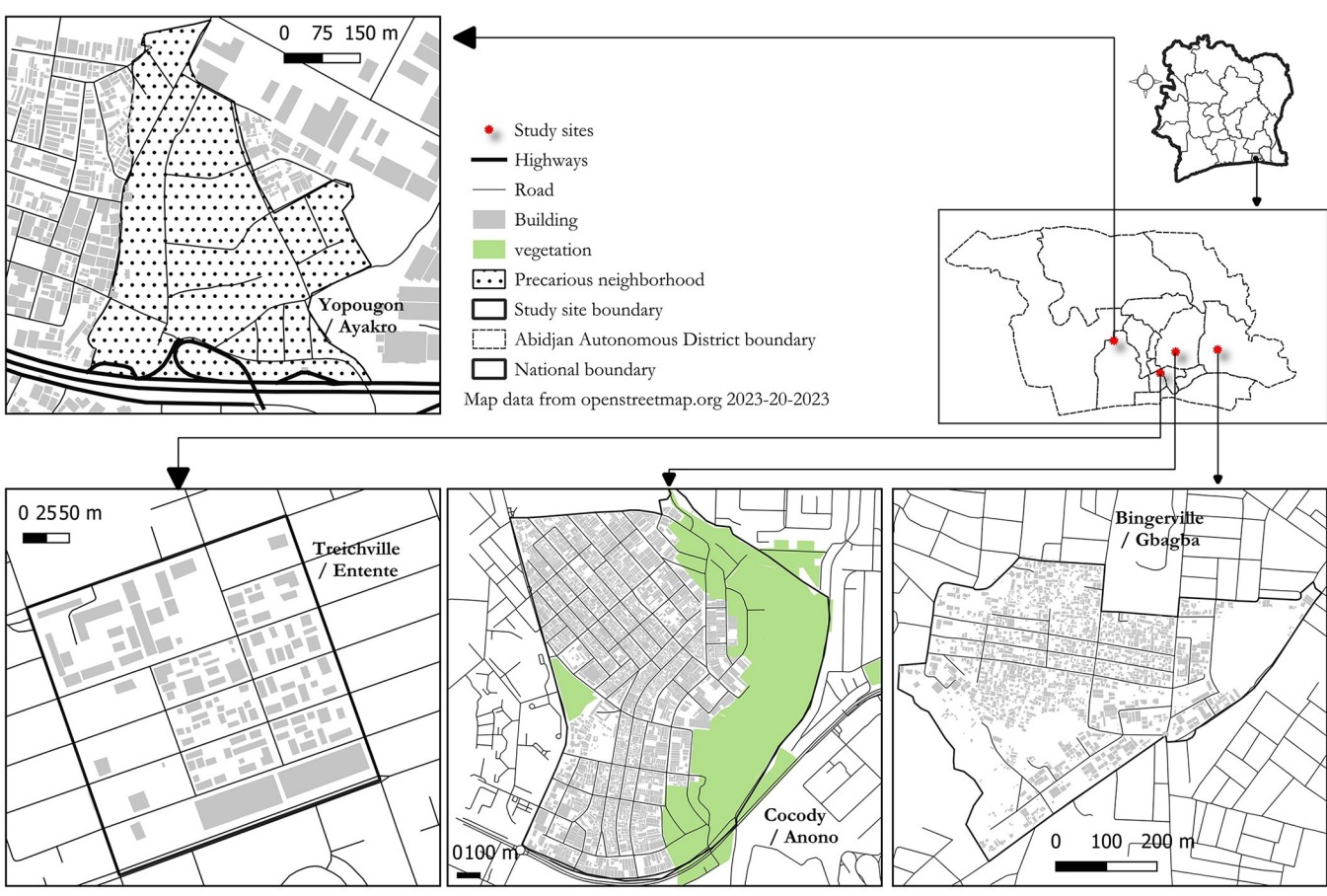

**Fig 1. Map showing the location of the study sites in the city of Abidjan, Côte d'Ivoire.** The study was conducted in four sites (Anono, Gbagba, Entente and Ayakro) in the city of Abidjan, Côte d'Ivoire. Anono, Gbagba, Entente and Ayakro located in the municipalities of Cocody, Bingerville, Treichville and Yopougon, respectively. As per arboviral epidemic reports from 2017 to 2023, Anono and Gbagba are situated in the health district of Cocody-Bingerville characterized by high arboviral incidence (80–90% dengue and yellow fever cases reported) while Ayakro and Entente are located in the health districts of Yopougon and Treichville characterized by low arboviral incidence (<10% dengue and yellow fever cases reported). The map was created with QGIS software version 3.34 (https://www.qgis.org/),using the basemap is Openstreetmap data so, the basemap is open data, licensed under the Open Data Commons Open Database License (ODbL) by the OpenStreetMap Foundation (OSMF).

## Study design

In the city of Abidjan, four sites of roughly equal size were selected based on their current arboviral status: Anono (Cocody), Gbagba (Bingerville), Entente (Treichville) and Ayakro (Yopougon). Anono and Gbagba are located in the health district of Cocody-Bingerville, where 80–90% of DEN and YF were recorded, while Ayakro and Entente are located in the health districts of Yopougon and Treichville, respectively, where only a small number of DEN and YF cases were recorded in the recent years (2017–2024). *Aedes* mosquito larvae were sampled among 100 households per study site and per survey. *Aedes* adults were collected from 10 houses in each study site for each survey. For each study site and each collection method, four surveys were carried out, corresponding each to one of the four seasons (i.e., SRS, LDS, LRS and SDS), from August 2019 to July 2020. The collections were done within and around randomly selected households and inside and outside of houses (I.e., main buildings). Surrounding areas within a 25-m radius around the selected households were investigated for *Aedes* larvae and adults. Household was defined as a house occupied by a head and his/her relatives.

If an occupant or owner of a selected household was absent or declined access to the house, this household was replaced by the nearest possible household.

### *Aedes* immature collections

*Aedes* mosquito immatures (larvae and pupae) were collected in visible and accessible water-holding containers indoors and outdoors among the selected houses. Any water-holding containers were inspected for the presence of *Aedes* immatures (i.e., larvae or pupae). Immatures of *Aedes* and non-*Aedes* (e.g., *Anopheles*, *Culex* and *Lutzia*) mosquitoes were collected using flexible rubber tube connected to a manual suction pump, ladles, and pipettes depending of the size of the breeding sites. Larvae of predatory mosquitoes (e.g., *Lutzia tigripes*) were removed from the samples to avoid predation on *Aedes* larvae or pupae. Mosquito immatures were filtered through a sieve. They were stored separately per breeding sites within the same water from the inspected larval breeding sites in plastic bags and labelled with information on the household number, study area and collection date. The larval breeding sites were characterized, recording their location (indoors or outdoors) and type (natural or artificial). The breeding sites were classified into five different categories: water storage containers, abandoned containers, tires, natural breeding site and others (e.g., hole of bricks, shoes, tarps, wooden boxes, mortars, metallic sheets). All mosquito immature samples were transferred into a cool box and transported to the insectarium for rearing to adults for morphological identification to species.

### *Aedes* adult collections

*Aedes* mosquito adults were captured using Prokopack aspirators (Model 1412, John W. Hock Company, USA). Sampling was performed by five well-trained and experienced entomological technicians per survey and the same technicians completed all the surveys. The technicians rotated from one survey to another to minimize biases. The collection box of each Prokopack aspirator was labelled with number of the households and houses sampled, and the initials of the collectors (i.e., technicians), start and end time of sampling, date of sampling and the study area were recorded. Sampling was carried out quarterly in 10 randomly selected houses in the domestic and peridomestic premises. In the domestic premises, sampling was carried out indoors and outdoors of the selected houses. Sampling was done in the morning from 06:00 a.m. to 09:00 a.m. and in the afternoon from 3:00 p.m. to 6:00 p.m. as *Aedes* are diurnal mosquitoes [20]. The time spent in each sampling point (i.e., house and surrounding area) was approximately 30 minutes.

### Laboratory procedures

In the insectarium, mosquito immatures were separated and placed up to 20 individuals per 200-ml plastic cup to prevent overcrowding and minimize morality during the rearing. The larvae were fed with fish food in the morning between 07:00 a.m. and 08:00 a.m. Emerged and field-collected pupae were kept until the emergence of adults. All emerged and field-collected adult mosquitoes were identified to species under a binocular magnifying glass using the morphological determination keys based on the color of the scutum, the appearance of the legs and wings, the shape of the thorax and the proboscis of the specimens [25–27]. Field-collected *Ae. aegypti* adult females' abdomen was examined, and females were classified as unfed, blood-fed, half-gravid and gravid individuals according to their abdomen aspect and blood-meal development stage [28]. The ovaries of field-collected *Ae. aegypti* females were dissected and the females were separated as parous or nulliparous individuals. Females were considered parous

when the ovary tracheoles were completely uncoiled, and nulliparous when the ovary tracheoles were coiled into pelota [29].

## Data analysis

All statistical analyses were conducted using R Studio version 4.2.0. Significance level of 5% was set for statistical testing. The proportion of positive breeding sites were calculated as the percentage of the number of breeding sites infested with *Ae. aegypti* larvae or pupae relative the total number of inspected water-holding breeding containers. The Z-test was used to compare the proportions of *Ae. aegypti*-positive breeding sites between the study sites and across the seasons.

The potential entomological risk of transmission of DEN and YF viruses was based on *Ae. aegypti* larval indices and assessed through the standard *Stegomyia* indices, including container index (CI), house index (HI) and Breteau index (BI). CI was equal to the percentage of *Ae. aegypti*-positive containers among the total number of water-holding containers inspected. A breeding site was considered positive when it contained at least one *Aedes* larva or pupa. HI was defined as the percentage of houses hosting at least one *Ae. aegypti*-positive container among the total number of households inspected. A house was considered positive when it contained at least one positive breeding site. BI was expressed as the number of *Ae. aegypti*-positive containers for 100 houses inspected. CI, HI and BI were compared between the four study sites and between seasons using generalized linear model (GLM to take into account possible interactions between the variables "study site", "season" or "breeding site" with poisson family.

The entomological risk of transmission of YF virus was interpreted according to the World Health Organization (WHO)-established YF epidemics thresholds [30]:

- if CI < 3%: epidemic risk is low; 3% ≤ CI ≤ 20%: risk is moderate, and CI > 20%: epidemic risk is high.

- if HI < 4%: risk of an epidemic is low. 4% ≤ HI ≤ 35%: risk of an epidemic is moderate. if HI > 35%: risk of an epidemic is high.

- if BI < 5: risk of an epidemic is low. 5 ≤ BI ≤ 50: risk is moderate. BI > 50: risk of an epidemic is high.

The entomological risk of transmission of DEN virus was defined and interpreted according to the Pan American Health Organization (PAHO)-established thresholds [31]:

- if CI > 3% or HI > 4% and BI > 5: risk of an epidemic is high.

- if HI < 0.1%, risk of an epidemic is low.

- if 0.1% ≤ HI ≤ 5%, the risk of an epidemic is medium.

- if HI > 5%, the risk of an epidemic is high.

For the field-collected *Ae. aegypti* adults, *Aedes* mean number was expressed as the number of *Aedes* adult specimens per house and per hour (AHH). AHH was tested using counting measure approaches in GLMs to consider possible interactions between the variables "study site", "season" or "house" with poisson family. When over-dispersion was found the negative binomial family structure was used. Repeated measures approach in GLM framework was used because *Aedes* mosquitoes were repeatedly sampled in the same sampling location (house) over time (season). The proportions of *Ae. aegypti* unfed, blood-fed, half-gravid and gravid females were calculated as the respective percentages of unfed, blood-fed, half-gravid

and gravid females (numerator) relative to the total number of females of the same species (denominator). The parity rate was defined as the percentage of parous females (numerator) among total number of females with ovaries dissected (denominator). The proportions of unfed, blood-fed, half-gravid and gravid females and parity rates were analyzed using the Z-test. The means of unfed, blood-fed, half-gravid and gravid female proportions and parity rates were compared between the study sites using GLM.

## Results

### Mosquito species composition

Table 1 shows the species composition of mosquitoes collected as immatures (larvae and pupae) and adults in Anono, Ayakro, Entente and Gbagba. A total of 86,796 mosquitoes (46,498 females and 40,298 males) was identified in all four study sites. Mosquitoes belonged to four genera (*Aedes* 97.15%, n = 84,319), *Culex* (2.68%, n = 2,322), *Anopheles* (0.16%, n = 136) and *Lutzia* (0.02%, n = 19), and seven species dominated by *Ae. aegypti* (97.14%, 84,317/86,796). *Culex quinquefasciatus* (67.16%, 1,965/2,926) was the most commonly found species in the adult collections, while *Ae. aegypti* represented 31.31% (916/2,926) of the field-collected adult mosquito fauna. *Ae. aegypti* (99.44%, 83,401/83,870) strongly dominated the culicid fauna obtained among the larval collections. The highest proportions of mosquito were recorded in Gbagba (30.43%, 26,414/86,796), followed by Ayakro (26.37%, 22,890/86,796), Anono (24.07%, 20,887/86,796) and Entente (19.13%, 16,605/86,796). *Aedes* genus dominated the mosquito fauna with overall proportion of 97.15% (n = 84,319), and in each study area with 98.11% (20,492/20,887) in Anono, 97.27% (16,152/16,605) in Entente, 97.15% (25,660/26,414) in Gbagba, and 96.18% (22,015/22,890) in Ayakro. *Ae. aegypti* was most abundant mosquito species in the four study sites (97.14%, n = 84,317). *Ae. aegypti* was the only *Aedes* species identified in the study sites, except for Gbagba where two additional specimens of *Aedes palpalis* species (0.01%, 2/25,658) were found. Other medically important non-*Aedes* species, such as *Culex quinquefasciatus* (2.65%, 2,298/86,796) a vector of arboviruses and *Anopheles gambiae s.l.* (0.16%, 136/86,796), a vector of *Plasmodium spp*, were also collected across the four study areas, though in relatively low proportions.

### *Aedes aegypti* immatures

**Breeding sites.** Table 2 displays the abundances of larval breeding sites of *Ae. aegypti* found in Anono, Ayakro, Entente and Gbagba across the seasons. In all the four study areas, a total 6,144 potential larval breeding containers were identified, with 21.5% (1,324/6,144) being positive for *Ae. aegypti* larvae. The proportions of *Ae. aegypti*-positive breeding sites varied from one site to another, with the highest proportions found in Gbagba (28.47%, 1,749/6,144), followed by Anono (26.03%, 1,599/6,144), Ayakro (25.65%, 1,576/6,144) and Entente (19.86%, 1,220/6,144). The proportions of *Ae. aegypti*-positive breeding sites were higher in the perido-mestic premises (31.9%, 424/1,328) compared with the domestic premises (18.7%, 900/4,816) (S1 Table). The domestic premises (900/1324, 68.0%) harbored higher numbers of *Ae. aegypti*-positive breeding sites than the peridomestic premises (424/1324, 32.0%). In the domestic premises, the majority of *Ae. aegypti*-positive larval breeding sites was found outdoors (19.5%, 851/4,360), while only small proportions were observed indoors (10.7%, 49/456) (S2 Table). Similarly, the numbers of *Ae. aegypti*-positive breeding sites were higher outdoors (851/900, 94.6%) than indoors (49/900, 5.4%).

GLMs indicated that *Ae. aegypti*-positive breeding sites were significantly different between seasons (F = 6.23, df = 3, p = 0.0009). The proportion of *Ae. aegypti* positive breeding sites did not differ statistically between LDS and SDS (Estimate = -0.42 ± 0.34, z = 1.21, p = 0.22).

**Table 1. Species composition of mosquitoes collected as immatures and adults in the study sites within the city of Abidjan, Côte d'Ivoire from August 2019 to July 2020.**

| Study site | Genus | Species | Immatures | | | | Adults | | | | Total | | | |
|---|---|---|---|---|---|---|---|---|---|---|---|---|---|---|
| | | | Female | Male | Total | % | Female | Male | Total | % | Female | Male | Total | % |
| **Anono** | *Aedes* | *Ae. aegypti* | 11119 | 9175 | 20294 | 99.79 | 81 | 117 | 198 | 35.93 | 11200 | 9292 | 20492 | 98.11 |
| | | **Sub-total** | **11119** | **9175** | **20294** | **99.79** | **81** | **117** | **198** | **35.93** | 11200 | 9292 | 20492 | **98.11** |
| | *Anopheles* | *An. gambiae s.l.* | 6 | 1 | 7 | 0.03 | 0 | 0 | 0 | 0.00 | 6 | 1 | 7 | 0.03 |
| | | ***Sub-total*** | **6** | **1** | **7** | **0.03** | **0** | **0** | **0** | **0.00** | 6 | 1 | 7 | 0.03 |
| | *Culex* | *Cx. quinquefasciatus* | 18 | 17 | 35 | 0.17 | 164 | 189 | 353 | 64.07 | 182 | 206 | 388 | 1.86 |
| | | **Sub-total** | **18** | **17** | **35** | **0.17** | **164** | **189** | **353** | **64.07** | 182 | 206 | 388 | 1.86 |
| | **Total** | | **11143** | **9193** | **20336** | **100** | **245** | **306** | **551** | **100** | **11388** | **9499** | **20887** | **100** |
| **Ayakro** | *Aedes* | *Ae. aegypti* | 11670 | 10137 | 21807 | 99.67 | 102 | 106 | 208 | 20.59 | 11772 | 10243 | 22015 | 96.18 |
| | | **Sub-total** | **11670** | **10137** | **21807** | **99.67** | **102** | **106** | **208** | **20.59** | 11772 | 10243 | 22015 | **96.18** |
| | *Anopheles* | *An. gambiae s.l.* | 28 | 24 | 52 | 0.24 | 12 | 3 | 15 | 1.49 | 40 | 27 | 67 | 0.29 |
| | | **Sub-total** | **28** | **24** | **52** | **0.24** | **12** | **3** | **15** | **1.49** | 40 | 27 | 67 | 0.29 |
| | *Culex* | *Cx. quinquefasciatus* | 10 | 9 | 19 | 0.09 | 408 | 379 | 787 | 77.92 | 418 | 388 | 806 | 3.52 |
| | | **Sub-total** | **10** | **9** | **19** | **0.09** | **408** | **379** | **787** | **77.92** | **418** | **388** | **806** | **3.52** |
| | *Lutzia* | *Lu. tigripes* | 0 | 2 | 2 | 0.01 | 0 | 0 | 0 | 0.00 | 0 | 2 | 2 | 0.01 |
| | | **Sub-total** | **0** | **2** | **2** | **0.01** | **0** | **0** | **0** | **0.00** | 0 | 2 | 2 | 0.01 |
| | **Total** | | **11708** | **10172** | **21880** | **100** | **522** | **488** | **1010** | **100** | **12230** | **10660** | **22890** | **100** |
| **Entente** | *Aedes* | *Ae. aegypti* | 8564 | 7336 | 15900 | 99.56 | 105 | 147 | 252 | 39.69 | 8669 | 7483 | 16152 | 97.27 |
| | | **Sub-total** | **8564** | **7336** | **15900** | **99.56** | **105** | **147** | **252** | **39.69** | 8669 | 7483 | 16152 | **97.27** |
| | *Anopheles* | *An. gambiae s.l.* | 18 | 15 | 33 | 0.21 | 0 | 0 | 0 | 0.00 | 18 | 15 | 33 | 0.20 |
| | | ***Sub-total*** | **18** | **15** | **33** | **0.21** | **0** | **0** | **0** | **0.00** | 18 | 15 | 33 | **0.20** |
| | *Culex* | *Cx. quinquefasciatus* | 16 | 11 | 27 | 0.17 | 157 | 226 | 383 | 60.31 | 173 | 237 | 410 | 2.47 |
| | | **Sub-total** | 16 | 11 | 27 | 0.17 | 157 | 226 | 383 | 60.31 | 173 | 237 | 410 | 2.47 |
| | *Lutzia* | *Lu. tigripes* | 7 | 3 | 10 | 0.06 | 0 | 0 | 0 | 0.00 | 7 | 3 | 10 | 0.06 |
| | | **Sub-total** | **7** | **3** | **10** | **0.06** | **0** | **0** | **0** | **0.00** | **7** | **3** | **10** | **0.06** |
| | **Total** | | **8605** | **7365** | **15970** | **100** | **262** | **373** | **635** | **100** | **8867** | **7738** | **16605** | **100** |
| **Gbagba** | *Aedes* | *Ae. aegypti* | 13491 | 11909 | 25400 | 98.89 | 104 | 154 | 258 | 35.34 | 13595 | 12063 | 25658 | 97.14 |
| | | *Ae. palpalis* | 0 | 0 | 0 | 0.00 | 1 | 1 | 2 | 0.27 | 1 | 1 | 2 | 0.01 |
| | | **Sub-total** | **13491** | **11909** | **25400** | **98.89** | **105** | **155** | **260** | **35.62** | 13596 | 12064 | 25660 | 97.15 |
| | *Anopheles* | *An. gambiae s.l.* | 11 | 14 | 25 | 0.10 | 3 | 1 | 4 | 0.55 | 14 | 15 | 29 | 0.11 |
| | | **Sub-total** | **11** | **14** | **25** | **0.10** | **3** | **1** | **4** | 0.55 | 14 | 15 | 29 | 0.11 |
| | *Culex* | *Cx. cinereus* | 0 | 0 | 0 | 0.00 | 7 | 4 | 11 | 1.51 | 7 | 4 | 11 | 0.04 |
| | | *Cx. nebulosus* | 0 | 0 | 0 | 0.00 | 7 | 6 | 13 | 1.78 | 7 | 6 | 13 | 0.05 |
| | | *Cx. quinquefasciatus* | 135 | 117 | 252 | 0.98 | 250 | 192 | 442 | 60.55 | 385 | 309 | 694 | 2.63 |
| | | **Sub-total** | **135** | **117** | **252** | **0.98** | **264** | **202** | **466** | **60.55** | 399 | 319 | 718 | 2.72 |
| | *Lutzia* | *Lu. tigripes* | 4 | 3 | 7 | 0.03 | 0 | 0 | 0 | 0.00 | 4 | 3 | 7 | 0.03 |
| | | **Sub-total** | **4** | **3** | **7** | **0.03** | **0** | **0** | **0** | **0.00** | 4 | 3 | 7 | 0.03 |
| | **Total** | | **13641** | **12043** | **25684** | **100** | **372** | **358** | **730** | **100** | **14013** | **12401** | **26414** | **100** |
| **Overall** | *Aedes* | *Ae. aegypti* | 44844 | 38557 | 83401 | 99.44 | 392 | 524 | 916 | 31.31 | 45236 | 39081 | 84317 | 97.14 |
| | | *Ae. palpalis* | 0 | 0 | 0 | 0.00 | 1 | 1 | 2 | 0.07 | 1 | 1 | 2 | 0.00 |
| | | **Sub-total** | **44844** | **38557** | **83401** | **99.44** | **393** | **525** | **918** | **31.37** | 45237 | 39082 | 84319 | **97.15** |
| | *Anopheles* | *An. gambiae s.l.* | 63 | 54 | 117 | 0.14 | 15 | 4 | 19 | 0.65 | 78 | 58 | 136 | 0.16 |
| | | **Sub-total** | **63** | **54** | **117** | **0.14** | **15** | **4** | **19** | **0.65** | **78** | **58** | **136** | **0.16** |
| | *Culex* | *Cx. cinereus* | 0 | 0 | 0 | 0.00 | 7 | 4 | 11 | 0.38 | 7 | 4 | 11 | 0.01 |
| | | *Cx. nebulosus* | 0 | 0 | 0 | 0.00 | 7 | 6 | 13 | 0.44 | 7 | 6 | 13 | 0.01 |
| | | *Cx. quinquefasciatus* | 179 | 154 | 333 | 0.40 | 979 | 986 | 1965 | 67.16 | 1158 | 1140 | 2298 | 2.65 |
| | | **Sub-total** | **179** | **154** | **333** | **0.40** | **993** | **996** | **1989** | **67.98** | 1172 | 1150 | 2322 | 2.68 |
| | *Lutzia* | *Lu. tigripes* | 11 | 8 | 19 | 0.02 | 0 | 0 | 0 | 0.00 | 11 | 8 | 19 | 0.02 |
| | | **Sub-total** | **11** | **8** | **19** | **0.42** | **0** | **0** | **0** | **0.00** | 11 | 8 | 19 | 0.02 |

*(Continued)*

**Table 1.** (Continued)

| Study site | Genus | Species | Immatures | | | | Adults | | | | Total | | | |
|---|---|---|---|---|---|---|---|---|---|---|---|---|---|---|
| | | | Female | Male | Total | % | Female | Male | Total | % | Female | Male | Total | % |
| **Total** | | | **45097** | **38773** | **83870** | **100** | **1401** | **1525** | **2926** | **100** | **46498** | **40298** | **86796** | **100** |

%: percentage. Immatures represent adult mosquitoes emerged from field-collected immatures (larvae and pupae). Adults represent adult field-collected mosquitoes.

However, *Aedes* breeding positivity was significantly higher in LDS compared with LRS (Estimate = 0.85 ± 0.23, z = 2.6, p = 0.0009) and in LDS compared with SRS (Estimate = 0.85 ± 0.32, z = 2.62, p = 0.008). The numbers of positive breeding sites were higher in Anono (22.4%, n = 358) than Entente (15.9%, n = 194) (Estimate = -0.80 ± 0.36, z = -2.23, p = 0.02), but no significant difference was found between Anono and Ayakro (Estimate = -0.05 ± 0.36, z = -0.14, p = 0.88), and Gbagba (Estimate = -0.06 ± 0.35, z = -0.19, p = 0.84). In Gbagba, the significantly lowest proportions of positive breeding sites of *Ae. aegypti* were observed in SDS (F = 3.98, df = 3, p = 0.03). Conversely, in the three other study sites, no significant differences in *Ae. aegypti*-positive breeding site was noticed between seasons (all p > 0.05). From all potential breeding sites, water storage containers (66.13%, n = 4,063) were the most predominant, followed by tires (20.56%, n = 1,263), discarded containers (8.74%, n = 537), other container categories (4.12%, n = 253) and natural breeding sites (0.46%, n = 28) ($\chi^2$ = 34.17, df = 4, p < 0.0001) (Table 2). Overall, *Ae. aegypti* colonized all the categories of breeding sites. *Aedes aegypti* larvae were found in water storage containers (15.01%, 610/4063), tires (31.67%, 400/1263), discarded containers (40.59%, 218/537), natural breeding sites (14.29%, 4/28) and other container categories (36.36%, 92/253).

On the 1,324 *Ae. aegypti*-positive breeding sites, water storage containers (46.07%, n = 610) were the most prevalent, followed by tires (30.21%, n = 400), discarded containers (16.46%, n = 218), the other container categories (6.95%, n = 92) and natural breeding sites (0.30%, n = 4) (Z-test $\chi^2$ = 1121.2, df = 4, p < 0.001). The positive breeding sites were found among other containers (4.2%, 15/358), discarded containers (9.8%, 35/358), water storage containers (29.9%, 107/358) and tires (56.1%, 201/358) in Anono. Water storage containers were found to be frequently positive in Ayakro (62.9%, 229/364), Entente (54.6%, 106/194) and Gbagba (41.2%, 168/408) seconded by tires in Ayakro (21.4%, 78/364) and discarded containers in Entente (19.6%, 38/194) and Gbagba (26.0%, 106/408).

In both Anono and Gbagba, the main *Aedes* breeding sites was discarded containers. In Anono, discarded containers were followed by the other container category (35.71%, 15/42), tires (28.47%, 201/706) and water storage containers (13.91%, 107/769), and tires (40.72%, 90/221), other containers (31.82%, 42/132) and water storage containers (14.65%, 168/1147) in Gbagba. In Ayakro, the main *Aedes* positive breeding sites were tires (45.88%, 78/170), followed by other containers (41.86%, 18/43) and discarded containers (38.61%, 39/101). In Entente, of the total *Aedes* breeding sites collected, other containers (47.22%, 17/36) were the most *Aedes*-positive breeding sites, followed by discarded containers (30.89%, 38/123).

In Anono, *Aedes*-positive breeding sites were mostly found in SRS (23.09%, 154/667), followed by LDS (22.84%, 66/289), LRS (22.65%, 106/468) and SDS (18.29%, 32/175) (Table 2). In contrast, in Ayakro, Gbagba and Entente, LRS had the most abundant *Aedes*-positive breeding sites, followed by SRS. The lowest proportion of *Aedes*-positive breeding sites was recorded in SDS in Gbagba (13.77%, 34/247) and Entente (7.34%, 13/177) and LDS in Ayakro (13.98%, 58/415).

**Immature productivity.** Table 3 presents the abundance of *Ae. aegypti* immatures (larvae and pupae) across the four study sites, seasons and larval breeding site categories. Out of

**Table 2. Seasonal variations of the abundances of the larval breeding sites of *Aedes aegypti* mosquitoes in the study sites within the city of Abidjan, Côte d'Ivoire from August 2019 to July 2020.**

| Study site | Breeding site | SRS | | | | LDS | | | | LRS | | | | SDS | | | | Total | | | |
|---|---|---|---|---|---|---|---|---|---|---|---|---|---|---|---|---|---|---|---|---|---|
| | | N | n | PW | PP | N | n | PW | PP | N | n | PW | PP | N | n | PW | PP | N | n | PW | PP |
| **Anono** | Water storage container | 315 | 58 | 18.4 | 37.7 | 128 | 14 | 10.9 | 21.2 | 226 | 29 | 12.8 | 27.4 | 100 | 6 | 6.0 | 18.8 | 769 | 107 | 13.9 | 29.9 |
| | Tire | 293 | 71 | 24.2 | 46.1 | 147 | 45 | 30.6 | 68.2 | 193 | 59 | 30.6 | 55.7 | 73 | 26 | 35.6 | 81.3 | 706 | 201 | 28.5 | 56.1 |
| | Discarded container | 42 | 18 | 42.9 | 11.7 | 6 | 3 | 50.0 | 4.5 | 33 | 14 | 42.4 | 13.2 | 1 | 0 | 0.0 | 0.0 | 82 | 35 | 42.7 | 9.8 |
| | Natural breeding site | 0 | 0 | na | 0.0 | 0 | 0 | na | 0.0 | 0 | 0 | na | 0.0 | 0 | 0 | na | 0.0 | 0 | 0 | na | 0.0 |
| | Others | 17 | 7 | 41.2 | 4.5 | 8 | 4 | 50.0 | 6.1 | 16 | 4 | 25.0 | 3.8 | 1 | 0 | 0.0 | 0.0 | 42 | 15 | 35.7 | 4.2 |
| | **Total** | **667** | **154** | **23.1** | **100.0** | **289** | **66** | **22.8** | **100.0** | **468** | **106** | **22.6** | **100.0** | **175** | **32** | **18.3** | **100.0** | **1599** | **358** | **22.4** | **100.0** |
| **Ayakro** | Water storage container | 384 | 89 | 23.2 | 61.0 | 366 | 43 | 11.7 | 74.1 | 331 | 66 | 19.9 | 55.9 | 180 | 31 | 17.2 | 73.8 | 1261 | 229 | 18.2 | 62.9 |
| | Tire | 58 | 32 | 55.2 | 21.9 | 28 | 7 | 25.0 | 12.1 | 59 | 31 | 52.5 | 26.3 | 25 | 8 | 32.0 | 19.0 | 170 | 78 | 45.9 | 21.4 |
| | Discarded container | 38 | 12 | 31.6 | 8.2 | 12 | 4 | 33.3 | 6.9 | 46 | 21 | 45.7 | 17.8 | 5 | 2 | 40.0 | 4.8 | 101 | 39 | 38.6 | 10.7 |
| | Natural breeding site | 1 | 0 | 0.0 | 0.0 | 0 | 0 | na | 0.0 | 0 | 0 | na | 0.0 | 0 | 0 | na | 0.0 | 1 | 0 | 0.0 | 0.0 |
| | Others | 27 | 13 | 48.1 | 8.9 | 9 | 4 | 44.4 | 6.9 | 5 | 0 | 0.0 | 0.0 | 2 | 1 | 50.0 | 2.4 | 43 | 18 | 41.9 | 4.9 |
| | **Total** | **508** | **146** | **28.7** | **100.0** | **415** | **58** | **14.0** | **100.0** | **441** | **118** | **26.8** | **100.0** | **212** | **42** | **19.8** | **100.0** | **1576** | **364** | **23.1** | **100.0** |
| **Entente** | Water storage container | 330 | 43 | 13.0 | 4.9 | 184 | 19 | 10.3 | 82.6 | 236 | 35 | 14.8 | 47.3 | 136 | 9 | 6.6 | 69.2 | 886 | 106 | 12.0 | 54.6 |
| | Tire | 63 | 11 | 17.5 | 13.1 | 4 | 2 | 50.0 | 8.7 | 72 | 16 | 22.2 | 21.6 | 27 | 2 | 7.4 | 15.4 | 166 | 31 | 18.7 | 16.0 |
| | Discarded container | 75 | 21 | 28.0 | 25.0 | 14 | 1 | 7.1 | 4.3 | 28 | 16 | 57.1 | 21.6 | 6 | 0 | 0.0 | 0.0 | 123 | 38 | 31.0 | 19.6 |
| | Natural breeding site | 4 | 1 | 25.0 | 1.19 | 3 | 0 | 0.0 | 0.0 | 1 | 0 | 0.0 | 0.0 | 1 | 1 | 100.0 | 7.7 | 9 | 2 | 22.2 | 1.0 |
| | Others | 13 | 8 | 61.5 | 9.5 | 4 | 1 | 25.0 | 4.3 | 12 | 7 | 58.3 | 9.5 | 7 | 1 | 14.3 | 7.7 | 36 | 17 | 47.2 | 8.8 |
| | **Total** | **485** | **84** | **17.3** | **100.0** | **209** | **23** | **11.0** | **100.0** | **349** | **74** | **21.2** | **100.0** | **177** | **13** | **7.3** | **100.0** | **1220** | **194** | **15.9** | **100.0** |
| **Gbagba** | Water storage container | 321 | 55 | 17.1 | 43.0 | 302 | 27 | 8.9 | 45.0 | 373 | 67 | 18.0 | 36.0 | 151 | 19 | 12.6 | 55.9 | 1147 | 168 | 14.6 | 41.2 |
| | Tire | 73 | 29 | 39.7 | 22.7 | 59 | 20 | 33.9 | 33.3 | 73 | 34 | 46.6 | 18.3 | 16 | 7 | 43.8 | 20.6 | 221 | 90 | 40.7 | 22.1 |
| | Discarded container | 71 | 33 | 46.5 | 25.8 | 23 | 11 | 47.8 | 18.3 | 115 | 56 | 48.7 | 30.1 | 22 | 6 | 27.3 | 17.6 | 231 | 106 | 45.9 | 26.0 |
| | Natural breeding site | 2 | 0 | 0.0 | 0.0 | 2 | 0 | 0.0 | 0.0 | 9 | 2 | 22.2 | 1.1 | 5 | 0 | 0.0 | 0.0 | 18 | 2 | 13.3 | 0.5 |
| | Others | 16 | 11 | 68.8 | 8.6 | 12 | 2 | 16.7 | 3.3 | 51 | 27 | 52.9 | 14.5 | 53 | 2 | 3.7 | 5.9 | 132 | 42 | 31.8 | 10.3 |
| | **Total** | **483** | **128** | **26.5** | **100.0** | **398** | **60** | **15.1** | **100.0** | **621** | **186** | **30.0** | **100.0** | **247** | **34** | **13.7** | **100.0** | **1749** | **408** | **23.3** | **100.0** |
| **Overall** | Water storage container | 1350 | 245 | 18.1 | 47.9 | 980 | 103 | 10.5 | 49.8 | 1166 | 197 | 16.9 | 40.7 | 567 | 65 | 11.5 | 53.7 | 4063 | 610 | 15.0 | 46.1 |
| | Tire | 487 | 143 | 29.4 | 27.9 | 238 | 74 | 31.1 | 35.7 | 397 | 140 | 35.3 | 28.9 | 141 | 43 | 30.5 | 35.5 | 1263 | 400 | 31.7 | 30.2 |
| | Discarded container | 226 | 84 | 37.2 | 16.4 | 55 | 19 | 34.5 | 9.2 | 222 | 107 | 48.2 | 22.1 | 34 | 8 | 23.5 | 6.6 | 537 | 218 | 40.6 | 16.5 |
| | Natural breeding site | 7 | 1 | 14.3 | 0.2 | 5 | 0 | 0.0 | 0.0 | 10 | 2 | 20.0 | 0.4 | 6 | 1 | 16.7 | 0.8 | 28 | 4 | 14.3 | 0.3 |
| | Others | 73 | 39 | 53.4 | 7.6 | 33 | 11 | 33.3 | 5.3 | 84 | 38 | 45.2 | 7.9 | 63 | 4 | 6.3 | 3.3 | 253 | 92 | 36.4 | 6.9 |
| | **Total** | **2143** | **512** | **23.9** | **100.0** | **1311** | **207** | **15.8** | **100.0** | **1879** | **484** | **25.8** | **100.0** | **811** | **121** | **14.9** | **100.0** | **6144** | **1324** | **21.5** | **100.0** |

N: number of wet containers inspected, n: *Aedes*-positive containers, PW: Percentage of *Aedes*-positive breeding sites among wet containers, PP: Proportion of each *Aedes*-positive breeding site type among the all *Aedes*-positive breeding site types. PW and PP are expressed as a percentage (%), na: not applicable, SRS: short rainy season, LDS: long dry season, LRS: long rainy season, SDS: short dry season. Others is the category of breeding containers composed of hole of brick, shoes, tarp, flower pot, wooden box, mortar, sheet metal. Natural breeding site is composed of water on land, leaf axils, snail shell, tree hole.

83,401 immatures of *Aedes*, 25,400 (30.5%) were found in Gbagba, 21,807 (26.1%) in Ayakro, 20,294 (24.3%) in Anono and 15,900 (19.1%) in Entente. GLM showed that no difference in the abundance of *Ae. aegypti* immatures among the four study sites (F = 0.64, df = 3, p = 0.59). The most productive larval habitats were water storage containers in SRS (43.7%, n = 12,611), tires in LDS (44.7%, n = 5,886) and LRS (33.7%, 12,213) and water storage containers in SDS (46.1%, n = 2,402). The abundance of *Ae. aegypti* immatures significantly varied across the seasons (F = 9.34, df = 3, p<0.0001). The highest number of *Ae. aegypti* immatures was collected in LRS (43.4%, n = 36,195) and the lowest in SDS (6.25%, n = 5,214).

Overall, the abundance of *Ae. aegypti* immature differed significantly according to the categories of the breeding sites (F = 6.14, df = 4, p = 0.0003). Of all breeding site categories, water storage containers (37.54%, n = 31,305) and tires (37.8%, n = 31,514) were those that produced more *Ae. aegypti* immatures, followed by discarded containers (19.4%, n = 16,200), others (4.6%, n = 3,798) and natural breeding sites (0.7%, n = 584). In all the four study sites, the main *Ae. aegypti* immature breeding sites were found in LRS and SRS. Conversely, the lowest number of *Ae. aegypti* immatures was found in both LDS and SDS. In Anono (F = 4.04, df = 3, p = 0.007), Gbagba (F = 7.01, df = 3, p < 0.001), Ayakro (F = 4.96, df = 3, p = 0.002) and Entente (F = 612.21, df = 3, p < 0.0001), the abundance of *Ae. aegypti* immatures showed significant difference across seasons (Table 3). The abundance of *Ae. aegypti* immatures during the LDS was significantly lower than during LRS (Estimates = 0.48 ± 0.21, z = 2.34, p = 0.01) and significantly higher than in SDS (Estimate = -0.54 ± 0.26, z = -2.07, p = 0.03). The abundance of *Ae. aegypti* immatures during the LDS was significantly higher than in SDS in Anono (Estimate = 0.79 ± 0.27, z = 2.91, p = 0.003) and in Entente (Estimate = -1.15 ± 0.55, z = -2.1, p = 0.03), but statistically lower than in LRS in Ayakro (Estimate = 0.86 ± 0.3, z = 2.88, p = 0.003) and Gbagba (Estimate = 0.79 ± 0.27, z = 2.9, p = 0.003).

### *Stegomyia* indices and dengue and yellow fever virus transmission risk

Table 4 shows the seasonal dynamics of *Stegomyia* indices and risk of transmission of DEN and YF viruses in Anono, Ayakro, Entente and Gbagba. Overall, CI, HI and BI were very high with respective values of 21.5%, 36.3% and 82.8, corresponding to the WHO density scale range of [6–7] and suggesting high risk of transmission of both DEN and YF. Gbagba, Ayakro and Anono were all at high risk, with respective WHO density scale ranges [6–8], [6–7] and [5–7], while entente was at the medium risk with the WHO density scale value of 5. The respective values of the overall CI, HI and BI were estimated at 23.3%, 43.0%, 102.0 in Gbagba, 23.1%, 43.5% and 91.0 in Ayakro, 22.4%, 33.5% and 89.5 in Anono, and 15.9%, 24.8% and 48.5 in Entente. In the all four study areas, CI, HI and BI were high and above the WHO-established epidemic thresholds for DEN. The overall CI, HI and BI were above the WHO-established epidemic thresholds for YF in Anono, Ayakro and Gbagba except for Entente, suggesting levels of risk of transmission of these arboviruses were high in Anono, Ayakro and Gbagba and moderate in Entente. Entente had significantly lower CI (Z-test $\chi^2$ = 14.08, df = 3, p = 0.002), HI (Z-test $\chi^2$ = 14.28, df = 3, p = 0.002) and BI (Z-test $\chi^2$ = 19.77, df = 3, p = 0.0001) compared with the three other study sites.

The overall risk of transmission of DEN and YF viruses varied from medium to high levels over the seasons and showed higher values during the rainy seasons, with the WHO density scale ranges of [7–8] in LRS (CI = 25.8%, HI = 47.8% and BI = 121.0) and [6–8] in SRS (CI = 23.9%, HI = 50.5% and BI = 128.0) (Table 4). CI, HI and BI showed seasonal variations all the study sites, as well. They all were higher than the WHO-established epidemic thresholds in any seasons for DEN and in LRS and SRS for YF (Fig 2). The overall CI values were higher during LRS (25.8%), followed by SRS (23.9%), LDS (15.8%) and SDS (14.9%) (Fig 2A). The

**Table 3. Seasonal variations of the abundances of *Aedes aegypti* immatures stages in the study sites within the city of Abidjan, Côte d'Ivoire from August 2019 to July 2020.**

| Study site | Breeding site | SRS | | LDS | | LRS | | SDS | | Total | |
|---|---|---|---|---|---|---|---|---|---|---|---|
| | | n | % | n | % | n | % | n | % | n | % |
| **Anono** | Water storage container | 2670 | 28.6 | 224 | 6.0 | 870 | 14.1 | 144 | 13.1 | 3908 | 19.3 |
| | Tire | 5182 | 55.6 | 3065 | 82.7 | 4228 | 68.6 | 954 | 86.9 | 13429 | 66.2 |
| | Discarded container | 1270 | 13.6 | 153 | 41.3 | 807 | 13.1 | 0 | 0.0 | 2230 | 11.0 |
| | Others | 206 | 2.2 | 262 | 7.1 | 259 | 4.2 | 0 | 0.0 | 727 | 3.6 |
| | **Total** | **9328** | **100.0** | **3704** | **100.0** | **6164** | **100.0** | **1098** | **100.0** | **20294** | **100.0** |
| **Ayakro** | Water storage container | 5134 | 63.2 | 1640 | 53.8 | 3916 | 44.6 | 1262 | 68.0 | 11952 | 54.8 |
| | Tire | 2332 | 28.7 | 720 | 23.6 | 3685 | 42.0 | 530 | 28.6 | 7267 | 33.3 |
| | Discarded container | 535 | 6.6 | 557 | 18.3 | 1176 | 13.4 | 53 | 2.9 | 2321 | 10.6 |
| | Others | 125 | 1.5 | 132 | 4.3 | 0 | 0.0 | 10 | 0.5 | 267 | 1.2 |
| | **Total** | **8126** | **100.0** | **3049** | **100.0** | **8777** | **100.0** | **1855** | **100.0** | **21807** | **100.0** |
| **Entente** | Water storage container | 2349 | 36.3 | 1332 | 57.0 | 2238 | 34.6 | 312 | 50.7 | 6231 | 39.2 |
| | Tire | 2385 | 36.8 | 671 | 28.7 | 2356 | 36.4 | 63 | 10.2 | 5475 | 34.4 |
| | Discarded container | 1176 | 18.1 | 92 | 3.9 | 1447 | 22.4 | 0 | 0.0 | 2715 | 17.1 |
| | Natural breeding site | 194 | 3.0 | 0 | 0.0 | 0 | 0.0 | 226 | 36.7 | 420 | 2.6 |
| | Others | 376 | 5.8 | 243 | 10.4 | 426 | 6.6 | 14 | 2.3 | 1059 | 6.7 |
| | **Total** | **6480** | **100.0** | **2338** | **100.0** | **6467** | **100.0** | **615** | **100.0** | **15900** | **100.0** |
| **Gbagba** | Water storage container | 2458 | 50.2 | 1707 | 42.0 | 4365 | 29.5 | 684 | 41.6 | 9214 | 36.3 |
| | Tire | 1430 | 29.2 | 1430 | 35.2 | 1944 | 13.1 | 539 | 32.7 | 5343 | 21.0 |
| | Discarded container | 840 | 17.1 | 837 | 20.6 | 6850 | 46.3 | 407 | 24.7 | 8934 | 35.2 |
| | Natural breeding site | 0 | 0.0 | 0 | 0.0 | 164 | 1.1 | 0 | 0.0 | 164 | 0.6 |
| | Others | 173 | 3.5 | 92 | 2.3 | 1464 | 9.9 | 16 | 1.0 | 1745 | 6.9 |
| | **Total** | **4901** | **100.0** | **4066** | **100.0** | **14787** | **100.0** | **1646** | **100.0** | **25400** | **100.0** |
| **Overall** | Water storage container | 12611 | 43.7 | 4903 | 37.3 | 11389 | 31.5 | 2402 | 46.1 | 31305 | 37.5 |
| | Tire | 11329 | 39.3 | 5886 | 44.7 | 12213 | 33.7 | 2086 | 40.0 | 31514 | 37.8 |
| | Discarded container | 3821 | 13.3 | 1639 | 12.5 | 10280 | 28.4 | 460 | 8.8 | 16200 | 19.4 |
| | Natural breeding site | 194 | 0.7 | 0 | 0.0 | 164 | 0.5 | 226 | 4.3 | 584 | 0.7 |
| | Others | 880 | 3.1 | 729 | 5.5 | 2149 | 5.9 | 40 | 0.8 | 3798 | 4.6 |
| | **Total** | **28835** | **100.0** | **13157** | **100.0** | **36195** | **100.0** | **5214** | **100.0** | **83401** | **100.0** |

%: percentage, n: number of larvae, SRS: short rainy season, LDS: long dry season, LRS: long rainy season, SDS: short dry season. Others is the category of breeding containers composed of hole of brick, shoes, tarp, flower pot, wooden box, mortar, sheet metal. Natural breeding site is composed of water on the ground, leaf axils, snail shell, tree hole.

highest CI values were recorded during LRS in Gbagba (30.0%), Entente (21.2%) and the whole study area (25.1%) and SRS in Ayakro (28.7%) and Anono (23.1%). The lowest CI values were found during LDS in Ayakro (14.0%) and SDS in Entente (7.3%), Gbagba (13.8%), Anono (18.3%) and the whole study area, (14.8%). The overall HI values were higher during SRS (50.5%), followed by LRS (47.8%), LDS (27.5%) and SDS (19.3%) (Fig 2B). The highest HI values were found during rainy seasons, especially during LRS in Gbagba (61.0%) and SRS in Ayakro (60.0%), Anono (48.0%), Entente (39.0%) and the whole study area (50.5%). All the lowest values of CI were observed during SDS throughout, with values of 11.0% in Entente, 15.0% in Anono, 25.0% in Gbagba and 26.0% in Ayakro and 19.3% in the whole study area. BI was higher during SRS (128.0), followed by LRS (121.0), LDS (51.8) and SDS (30.3) in the whole study site (Fig 2C). The highest BI values were recorded during the rainy seasons, LRS in Gbagba (186.0) and SRS in Anono (154.0), Ayakro (146.0), Entente (84.0) and the whole study area (128.0). The lowest BI were observed during SDS in all the study sites, showing

**Table 4. Risk of transmission of dengue and yellow fever viruses in the study sites within the city of Abidjan, Côte d'Ivoire from August 2019 to July 2020.**

| Study site | Season | CI (%) | HI (%) | BI | WHO density scale | Risk level | |
|---|---|---|---|---|---|---|---|
| | | | | | | Dengue | Yellow fever |
| **Anono** | SRS | 23.1 | 48.0 | 154.0 | 6–8 | High | High |
| | LDS | 22.8 | 27.0 | 66.0 | 5–6 | High | Medium |
| | LRS | 22.7 | 44.0 | 106.0 | 6–8 | High | High |
| | SDS | 18.3 | 15.0 | 32.0 | 4–5 | High | Medium |
| | **Total** | **22.4** | **33.5** | **89.5** | **5–7** | **High** | **Medium** |
| **Ayakro** | SRS | 28.7 | 60.0 | 146.0 | 7–8 | High | High |
| | LDS | 14.0 | 34.0 | 58.0 | 4–6 | High | Medium |
| | LRS | 26.8 | 55.0 | 118.0 | 7–8 | High | High |
| | SDS | 19.8 | 26.0 | 42.0 | 5 | High | Medium |
| | **Total** | **23.1** | **43.5** | **91.0** | **6–7** | **High** | **High** |
| **Entente** | SRS | 17.3 | 39.0 | 84.0 | 5–7 | High | High |
| | LDS | 11.0 | 18.0 | 23.0 | 4 | High | Medium |
| | LRS | 21.2 | 31.0 | 74.0 | 5–6 | High | Medium |
| | SDS | 7.3 | 11.0 | 13.0 | 3 | High | Medium |
| | **Total** | **15.9** | **24.8** | **48.5** | **5** | **High** | **Medium** |
| **Gbagba** | SRS | 26.5 | 55.0 | 128.0 | 8 | High | High |
| | LDS | 15.1 | 31.0 | 60.0 | 5–6 | High | Medium |
| | LRS | 30.0 | 61.0 | 186.0 | 8 | High | High |
| | SDS | 13.8 | 25.0 | 34.0 | 4–5 | High | Medium |
| | **Total** | **23.3** | **43.0** | **102.0** | **6–8** | **High** | **High** |
| **Overall** | SRS | 23.9 | 50.5 | 128.0 | 6–8 | High | High |
| | LDS | 15.8 | 27.5 | 51.8 | 5–6 | High | Medium |
| | LRS | 25.8 | 47.8 | 121.0 | 7–8 | High | High |
| | SDS | 14.9 | 19.3 | 30.3 | 4 | High | Medium |
| | **Total** | **21.5** | **36.3** | **82.8** | **6–7** | **High** | **High** |

%: percentage, CI: Container index, HI: House index, BI: Breteau index, SRS: short rainy season, LDS: long dry season, LRS: long rainy season, SDS: short dry season, WHO: World Health Organization, Risk levels estimated according to WHO [30, 31]

values of 13.0% in Entente, 32.0% in Anono, 34.0% in Gbagba and 42.0% in Ayakro and 30.3% in the whole study area.

### *Aedes aegypti* adults

**Abundance.** In total, 916 *Ae. aegypti* adults were identified, with higher proportion in Gbagba (28.17%), followed by Entente (27.51%), Ayakro (22.71%), and Anono (21.62%). *Aedes aegypti* populations were composed of 40.91% females and 59.09% males in Anono (n = 198), 40.31% females and 59.69% males in Gbagba (n = 258), 50.96% females and 49.04% males in Ayakro (n = 208) and 41.67% females and 58.33% males in Entente (n = 252).

Overall AHH (mean ± standard error) was of 4.95 ± 0.86 *Aedes*/house/hour in Anono, 5.2 ± 1.06 *Aedes*/house/hour in Ayakro, 6.30 ± 1.32 *Aedes*/house/hour in Entente, and 6.45 ± 1.07 *Aedes*/house/hour in Gbagba (Fig 3). The respective AHHs of females were of 2.02 ± 0.33, 2.6 ± 0.38, 2.65 ± 0.68 and 2.62 ± 0.60 female/house/hour in Anono, Ayakro and Entente and Gbagba (Table 5). Males' AHH was 3.85 ± 0.76, 3.68 ± 0.82, 2.92 ± 0.57 and, 2.55 ± 0.46, male/house/hour in Gbagba, Entente, Anono and Ayakro, respectively (Table 5). AHH did not differ significantly between the four study sites for *Ae. aegypti* overall

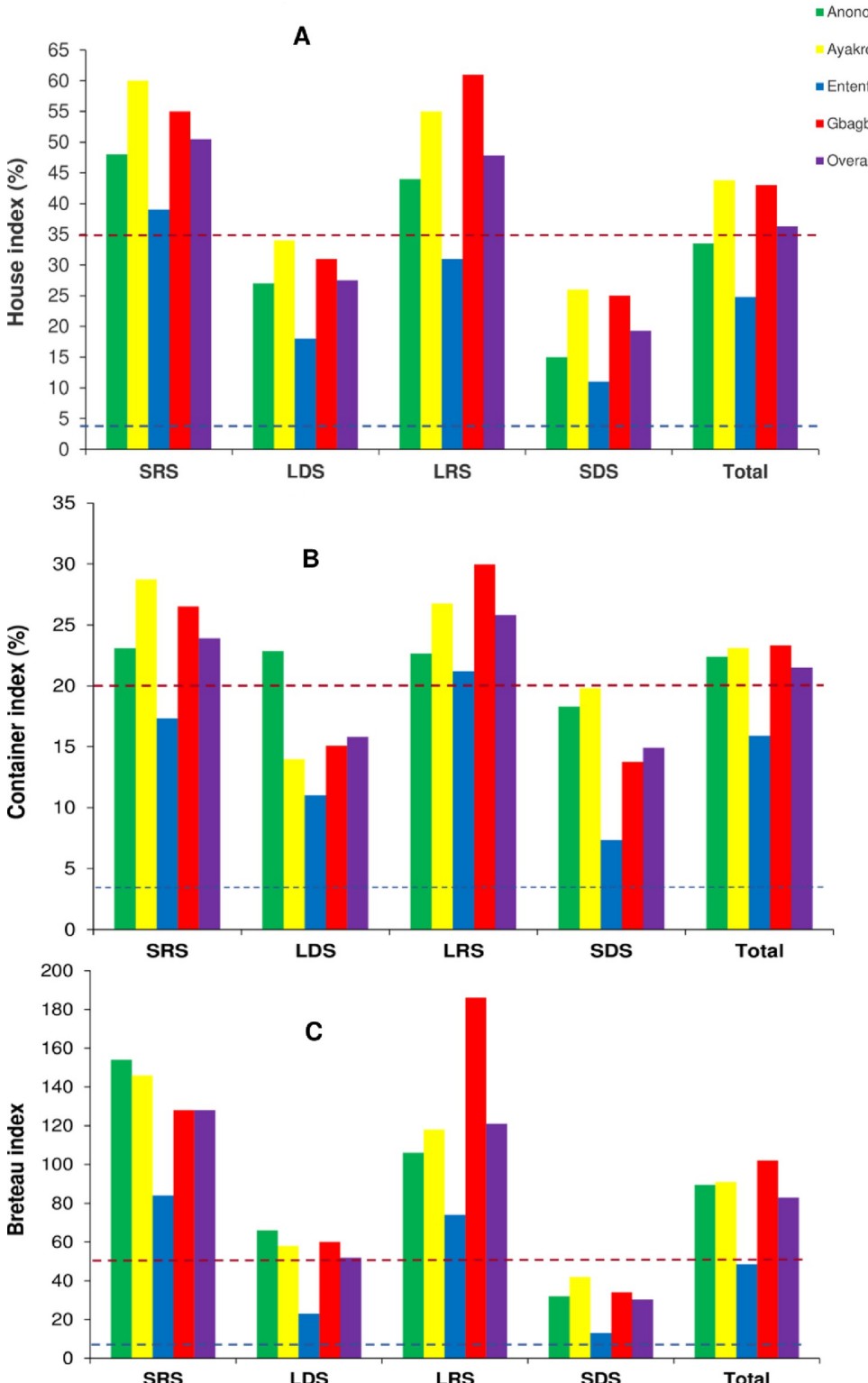

**Fig 2. Seasonal variations of *Aedes aegypti* immature indices in the study sites within the city of Abidjan, Côte d'Ivoire.** A: Container index (CI), B: House index (HI), C: Breteau index (BI). The blue dotted line represents the dengue epidemic threshold levels and the red dotted line represents the yellow fever epidemic threshold levels. The dengue epidemic threshold levels are 3% for container index, 4% for house index and 5 for Breteau index [31]. The yellow fever epidemic threshold levels are 20% for container index, 35% for house index and 50 for Breteau index [30]. SRS: short rainy season, LDS: long dry season, LRS: long rainy season, SDS: short dry season.

populations (F = 0.49, df = 3, p = 0.68), for females (F = 0.38, df = 3, p = 0.76) and for males (F = 0.92, df = 3 p = 0.43).

AHHs of *Ae. aegypti* varied significantly in the four study sites across the seasons (F = 21.79, df = 3, p < 0.001) (Fig 3). The highest AHHs were observed during SRS in Entente (12.3 ± 3.94 *Aedes*/house/hour), followed by Gbagba (10.2 ± 2.5 *Aedes*/house/hour) and Anono (8.6 ± 1.8 *Aedes*/house/hour), except for Ayakro where the maximum AHH (7.5 ± 1.71 *Aedes*/house/hour) was recorded in LRS. Conversely, the lowest AHHs were recorded during SDS in Anono (1.7 ± 0.60 *Aedes*/house/hour) and Gbagba (2.8 ± 0.68 *Aedes*/house/hour) and LDS in Ayakro (2.4 ± 0.78 *Aedes*/house/hour) and Entente (2.5 ± 0.58 *Aedes*/house/hour). AHHs were not statistically different between the four study sites during the same season, SRS (F = 0.46, df = 3, p = 0.71), LDS (F = 0.83, df = 3, p = 0.48), LRS (F = 0.16, df = 3, p = 0.92) (Fig 3). AHHs did not show any statistical differences between the four study sites in the same seasons, nor in the females and neither the males. In contrast, AHHs were significantly different over seasonal variations between the four study sites for *Ae. aegypti* females (F = 13.77, df = 3, p < 0.001) and males (F = 21.12, df = 3, p < 0.001).

## Resting behaviors

*Aedes aegypti* was highly exophilic, with 93.4% (856/916) of individuals collected outdoors of houses (S3 Table). Only 6.6% of individuals was collected indoors of houses. The overall proportions of exophilic individuals varied slightly among the four study areas, varying between 90.4% (188/208) and 95.5% (189/198). Similarly, the proportions of exophilic individuals varied little across the seasons in all the study sites. *Aedes aegypti* showed lowest proportions of 78.7% (59/75) in Ayakro during LRS and highest proportions of 100% (19/19) in Anono during LDR outdoors of houses.

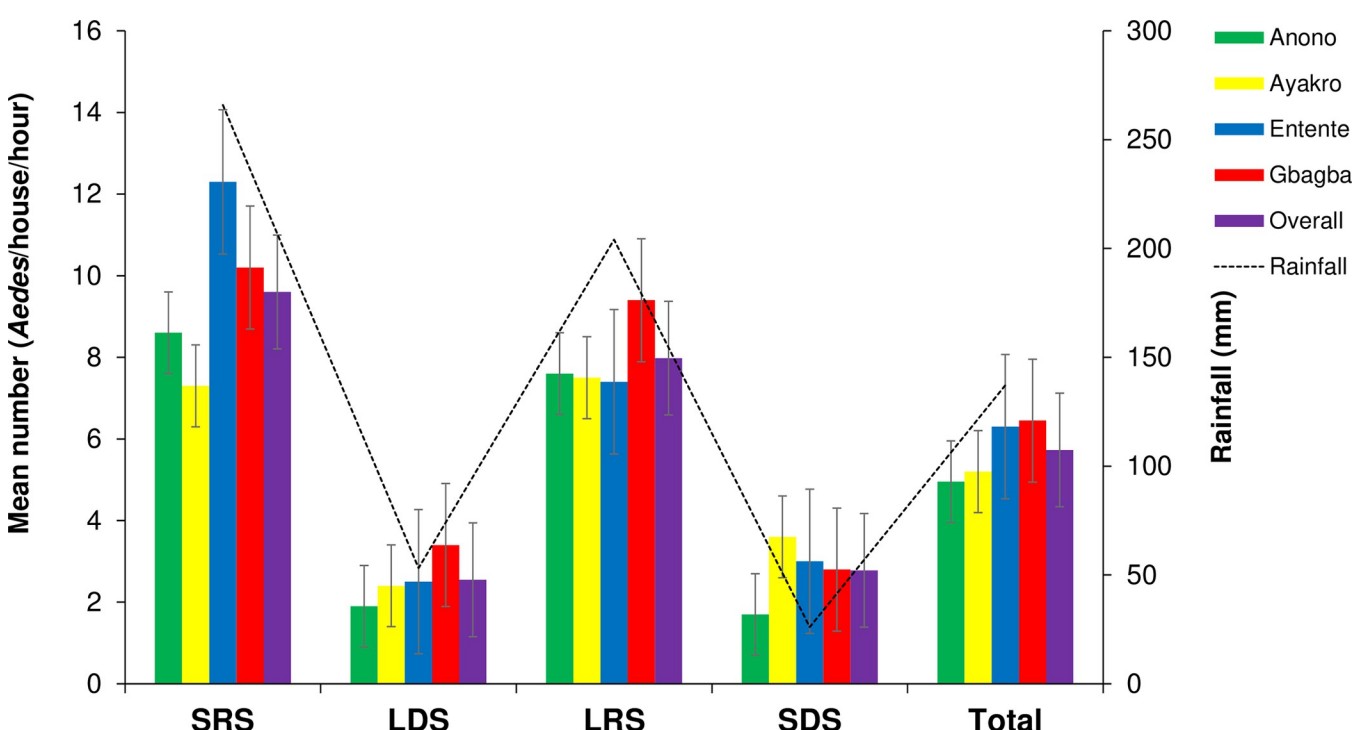

**Fig 3. Seasonal abundance of the adult populations of *Aedes aegypti* in the study sites within the city of Abidjan, Côte d'Ivoire.** SRS: short rainy season, LDS: long dry season, LRS: long rainy season, SDS: short dry season. Error bars show the standard error (SE).

**Table 5. Seasonal variations of the abundance of *Aedes aegypti* adult populations in the study sites within city of Abidjan, Côte d'Ivoire from August 2019 to July 2020.**

| Study site | Indicator | SRS | LDS | LRS | SDS | Total |
|---|---|---|---|---|---|---|
| **Anono** | Number | 31 | 10 | 30 | 10 | 81 |
| | Proportion (%) | 38.3 | 12.3 | 37.0 | 12.3 | 100 |
| | AHH (mean ± SE) (female + male) | 8.6 ± 1.80 | 1.9 ± 0.52 | 7.6 ± 2.13 | 1.7 ± 0.59 | 4.95 ± 0.86 |
| | AHH (mean ± SE) (female) | 3.10 ± 0.71 | 1.0 ± 0.33 | 3.0 ± 0.83 | 1.0 ± 0.33 | 2.02 ± 0.33 |
| | AHH (mean ± SE) (male) | 5.5 ± 1.2 | 0.9 ± 0.27 | 4.6 ± 1.44 | 0.7 ± 0.3 | 2.92 ± 0.57 |
| **Ayakro** | Number | 39 | 12 | 41 | 14 | 106 |
| | Proportion (%) | 36.8 | 11.3 | 38.7 | 13.2 | 100 |
| | AHH (mean ± SE) (female + male) | 7.3 ± 3.54 | 2.4 ± 0.77 | 7.5 ± 1.71 | 3.6 ± 1.10 | 5.2 ± 1.06 |
| | AHH (mean ± SE) (female) | 3.90 ± 2.50 | 1.20 ± 0.50 | 4.10 ± 0.70 | 1.40 ± 0.43 | 2.65 ± 0.68 |
| | AHH (mean ± SE) (male) | 3.4 ± 1.17 | 1.2 ± 0.38 | 3.4 ± 1.13 | 2.2 ± 0.69 | 2.55 ± 0.45 |
| **Entente** | Number | 49 | 13 | 32 | 11 | 105 |
| | Proportion (%) | 46.7 | 12.4 | 30.5 | 10.5 | 100 |
| | AHH (mean ± SE) (female + male) | 12.3 ± 3.94 | 7.5 ± 0.58 | 7.4 ± 2.52 | 3.0 ± 0.97 | 6.3 ± 1.32 |
| | AHH (mean ± SE) (female) | 4.9 ± 1.75 | 1.30 ± 0.34 | 3.20 ± 1.36 | 1.10 ± 0.41 | 2.62 ± 0.60 |
| | AHH (mean ± SE) (male) | 7.4 ± 2.58 | 1.20 ± 0.32 | 4.20 ± 1.34 | 1.9 ± 0.64 | 3.68 ± 0.81 |
| **Gbagba** | Number | 34 | 15 | 42 | 13 | 104 |
| | Proportion (%) | 32.7 | 14.4 | 40.4 | 12.5 | 100 |
| | AHH (mean ± SE) (female + male) | 10.2 ± 2.50 | 3.4 ± 0.77 | 9.4 ± 2.75 | 2.8 ± 0.68 | 6.45 ± 1.07 |
| | AHH (mean ± SE) (female) | 3.40 ± 0.85 | 1.50 ± 0.40 | 4.20 ± 0.93 | 1.30 ± 0.40 | 2.60 ± 0.39 |
| | AHH (mean ± SE) (male) | 6.8 ± 1.93 | 1.9 ± 0.58 | 5.2 ± 1.86 | 1.5 ± 0.40 | 3.85 ± 0.75 |
| **Overall** | Number | 153 | 50 | 145 | 48 | 396 |
| | Proportion (%) | 38.6 | 12.6 | 36.6 | 12.1 | 100.0 |
| | AHH (mean ± SE) (female + male) | 9.6 ± 1.50 | 2.55 ± 0.33 | 7.98 ± 1.12 | 2.78 ± 0.42 | 5.72 ± 0.54 |
| | AHH (mean ± SE) (female) | 3.82 ± 0.78 | 1.25 ± 0.19 | 3.62 ± 0.47 | 1.20 ± 0.19 | 2.48 ± 0.26 |
| | AHH (mean ± SE) (male) | 5.78 ± 0.90 | 1.3 ± 0.20 | 4.35 ± 0.71 | 1.58 ± 0.27 | 3.25 ± 0.33 |

% percentage, AHH: *Aedes* adult per house per hour, SE: standard error, SRS: short rainy season, LDS: long dry season, LRS: long rainy season, SDS: short dry season.

## Blood-meal development status

Fig 4 indicates the seasonal variations of the blood-meal status of female *Ae. aegypti* adult populations in Anono, Ayakro, Entente and Gbagba. Overall, *Ae. aegypti* females were mostly unfed (51.3 ± 2.5%, 203/396), followed by blood-fed (22.2 ± 2.1%, 88/396), gravid (13.9 ± 1.7%, 55/396) and half-gravid (12.6 ± 1.7%, 50/396). The highest proportions of unfed females were found in Entente (63.8 ± 4.7%, 67/105), followed by Ayakro (48.1 ± 4.9%, 51/106), Gbagba (47.1 ± 4.9%, 49/104) and Anono (44.4 ± 5.5%, 81/106) (Fig 4 and S4 Table). The proportions of blood-fed females were higher in Ayakro (28.3 ± 4.4%, 30/106), followed by Anono (25.9 ± 4.9%, 21/81), Gbagba (19.2 ± 3.9%, 20/104) and Entente (16.2 ± 3.6%, 17/105). The lowest proportions belonged to half-gravid females in Anono (11.1 ± 3.5%, n = 81), Ayakro (9.43 ± 2.84, n = 106), Entente (7.62 ± 2.59%, n = 105) and gravid females in Gbagba (16.35 ± 3.63, n = 104). The proportions of blood-fed (Z-test $\chi^2$ = 5.65, df = 3, p = 0.12), half-gravid (Z-test $\chi^2$ = 5.5, df = 3, p = 0.13) and gravid (Z-test $\chi^2$ = 1.53, df = 3, p = 0.67) females were statistically similar between the four study sites. However, the proportion of unfed females was significantly higher in Entente than in the three other study sites (Z-test $\chi^2$ = 9.26, df = 3, p = 0.020).

In general, the proportions of unfed females were higher during the rainy seasons (57.2 ± 4.1% in LRS and 52.9 ± 4.0% in SRS) compared with the dry seasons (42.0 ± 7.0% in LDS and 37.5 ± 7.0% in SDS) (S2 Fig). In contrast, the proportions of blood-fed females were

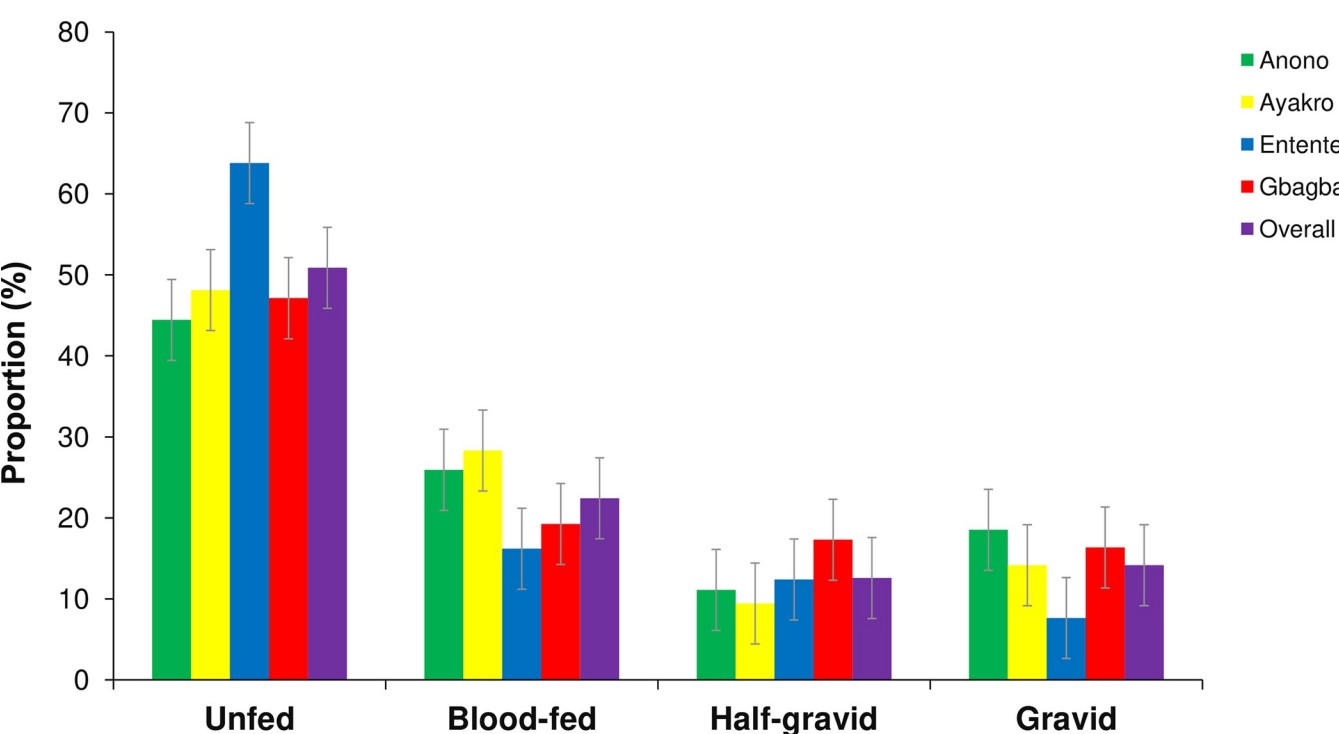

**Fig 4. Blood-meal statuses in adult females of *Aedes aegypti* in the study sites within the city of Abidjan, Côte d'Ivoire.** Error bars show the standard error (SE).

higher during the dry seasons (30.0 ± 6.5% in LDS and 27.1 ± 6.4% in SDS) than during the rainy seasons (22.8 ± 3.5% in LRS and 17.6 ± 3.1% in SRS). Generally, the seasonality did not significantly influence the proportions of blood-fed females in all the four study sites (Z-test $\chi^2$ = 4.28, df = 3, p = 0.23). Moreover, the proportions of unfed, half-gravid, and gravid females were each comparable among the seasons in each study site (all p > 0.05) (S3, S4, S5 and S6 Figs).

## Parity

Table 6 displays the seasonal variations of the parity rate of *Ae. aegypti* populations in Anono, Ayakro, Entente and Gbagba. *Ae. aegypti* parity rate varied across study sites, with rates recorded at 48,1% in Anono, 51,9% in Ayakro, 48,6% in Entente, and 50,0% in Gbagba. The parity rate did not significantly differ between the four study sites (Z-test $\chi^2$ = 0.33, df = 3, p = 0.95). In addition, there was no significant difference in the proportions of parous and nulliparous females throughout (Z- test $\chi^2$ = 0.06, df = 1, p = 0.802). Overall, the parity rate showed significant difference across seasonal variations (Z-test $\chi^2$ = 20.95, df = 3, p = 0.0001). Indeed, the highest parity rates were found during SRS in Ayakro (64.1 ± 7.7%) and Anono (61.3 ± 8.7%) and during LDS in Entente (69.2 ± 12.8%) and Gbagba (66.7 ± 12.2) (S7 Fig). The parity rate was not significantly different across seasonal variations in Anono (Z-test $\chi^2$ = 4.0, df = 3, p = 0.26) and Entente (Z-test $\chi^2$ = 6.07, df = 3, p = 0.10), but showed significant difference over the seasons in Gbagba (Z-test $\chi^2$ = 8.29, df = 3, p = 0.04) and Ayakro (Z-test $\chi^2$ = 9.65, df = 3, p = 0.02).

## Discussion

As most sub-Saharan African cities, the highly urbanized city of Abidjan, Côte d'Ivoire has faced multiple outbreaks of DEN coupled with YF cases from 2017 to 2024 [13–18,32–35].

**Table 6. Seasonal variations of the abundance and the parity status of female populations of *Aedes aegypti* in the study sites within the city of Abidjan, Côte d'Ivoire from August 2019 to July 2020.**

| Study site | Indicator | | SRS | LDS | LRS | SDS | Total |
|---|---|---|---|---|---|---|---|
| **Anono** | Abundance | Number | 31 | 10 | 30 | 10 | 81 |
| | | Proportion (%) | 38.3 | 12.3 | 37.0 | 12.3 | 100 |
| | | AHH (mean ± SE) | 3.10 ± 0.71 | 1.0 ± 0.33 | 3.0 ± 0.83 | 1.0 ± 0.33 | 2.02±0.33 |
| | Parity | Parous | 19 | 5 | 11 | 4 | 39 |
| | | Nulliparous | 12 | 5 | 19 | 6 | 42 |
| | | Parity rate (mean ± SE) (%) | 61.3 ± 8.7 | 50.0 ± 15.8 | 36.7 ± 8.8 | 40.0 ± 15.5 | 48.1±5.5 |
| **Ayakro** | Abundance | Number | 39 | 12 | 41 | 14 | 106 |
| | | Proportion (%) | 36.8 | 11.3 | 38.7 | 13.2 | 100 |
| | | AHH (mean ± SE) | 3.90 ± 2.50 | 1.20 ± 0.50 | 4.10 ± 0.70 | 1.40 ± 0.43 | 2.65±0.68 |
| | Parity | Parous | 25 | 6 | 18 | 6 | 55 |
| | | Nulliparous | 14 | 6 | 23 | 8 | 51 |
| | | Parity rate (mean ± SE) (%) | 64.1 ± 7.7 | 50.0 ± 14.4 | 43.9 ± 7.7 | 42.9 ± 13.2 | 51.9±4.9 |
| **Entente** | Abundance | Number | 49 | 13 | 32 | 11 | 105 |
| | | Proportion (%) | 46.7 | 12.4 | 30.5 | 10.5 | 100 |
| | | AHH (mean ± SE) | 4.90 ± 1.75 | 1.30 ± 0.34 | 3.20 ± 1.36 | 1.10 ± 0.41 | 2.62±0.60 |
| | Parity | Parous | 19 | 9 | 16 | 7 | 51 |
| | | Nulliparous | 30 | 4 | 16 | 4 | |
| | | Parity rate (mean ± SE) (%) | 38.8 ± 7.0 | 69.2 ± 12.8 | 50.0 ± 8.8 | 63.6 ± 14.5 | 48.6±4.9 |
| **Gbagba** | Abundance | Number | 34 | 15 | 42 | 13 | 104 |
| | | Proportion (%) | 32.7 | 14.4 | 40.4 | 12.5 | 100 |
| | | AHH (mean ± SE) | 3.40 ± 0.85 | 1.50 ± 0.40 | 4.20 ± 0.93 | 1.30 ± 0.40 | 2.60 ± 0.39 |
| | Parity | Parous | 21 | 10 | 14 | 7 | 52 |
| | | Nulliparous | 13 | 5 | 28 | 6 | 48 |
| | | Parity rate (mean ± SE) (%) | 61.8 ± 8.3 | 66.7 ± 12.2 | 33.3 ± 7.3 | 53.8 ± 13.8 | 50.0 ± 4.9 |
| **Overall** | Abundance | Number | 153 | 50 | 145 | 48 | 396 |
| | | Proportion (%) | 38.6 | 12.6 | 36.6 | 12.1 | 100.0 |
| | | AHH (mean ± SE) | 3.82 ± 0.78 | 1.25 ± 0.19 | 3.62 ± 0.47 | 1.20 ± 0.19 | 2.48 ± 0.26 |
| | Parity | Parous | 84 | 30 | 59 | 24 | 197 |
| | | Nulliparous | 69 | 20 | 86 | 24 | 199 |
| | | Parity rate (mean ± SE) (%) | 54.9 ± 4.0 | 60.0 ± 6.9 | 40.7 ± 4.1 | 50.0 ± 7.2 | 49.7 ± 2.5 |

%: percentage, AHH: *Aedes* adult per house per house, SE: standard error, SRS: short rainy season, LDS: long dry season, LRS: long rainy season, SDS: short dry season.

Therefore, it is crucial to better understand the extent to which *Ae. aegypti* and arbovirus-related epidemic risks spread across Abidjan. Thus, this study assessed and compared the ecology of *Ae. aegypti* and the risk of transmission of DEN and YF viruses in Abidjan, among Anono, Ayakro, Gbagba and Entente characterized by geographical differences in reported arboviral incidences. Anono and Gbagba are located in the health district of Cocody-Bingerville that accounted for 80–90% reported cases of DEN and YF, while only few cases of arboviral diseases (<10%) were recorded in Ayakro and Entente. The results showed that all the four study sites were heavily infested with *Ae. aegypti*, resulting in medium to high levels of risk of DEN and YF virus transmission. Overall, no significant differences were observed in the ecological patterns of local *Ae. aegypti* populations, and *Stegomyia* indices (CI: 21.2%, HI: 36.3% and BI: 82.8) found to be higher than the WHO-established epidemic thresholds. These findings suggest that the populations are exposed to very high and similar threats from DEN and YF outbreaks in the study sites, potentially across the large city of Abidjan, Côte d'Ivoire.

Our data showed that *Ae. aegypti*, the main arbovirus vector [8], was the most abundant mosquito species (>97%) and almost the only *Aedes* species in all the study sites. The dominance of *Ae. aegypti* found in our study sites is consistent with that recorded traditionally in large African urbanized cities where *Ae. albopictus* are absent or rare, as reported in Abidjan [36] and Ouagadougou, Burkina Faso [36]. Its high abundance among larval collections may be explained by the fact that this study was conducted in the urban area of Abidjan. The dominance of this vector in urban areas of Africa is well documented in several studies [37,38]. Similarly, its predominance among the mosquitoes collected during the larval survey can also be explained by the types of containers inspected (domestic or abandoned containers, tires, etc.). These are preferential breeding sites for this vector [39,24]. *Aedes albopictus*, another key arbovirus vector species previously notified in Abidjan in 2010 [40] and 2014 [19], was not sampled in our present study. Although the presence of *Ae. albopictus* has been reported in Abidjan, its absence in our mosquito samples could be attributed to its establishment in an environment where *Ae. aegypti* was already well established, which may have led to its gradual disappearance. According to Hashim et al. [41], to avoid competition, *Ae. aegypti* tends not to lay its eggs in sites already colonized by *Ae. aegypti* and vice versa. This indicates that these two species have difficulties cohabiting in the same breeding site [41]. Moreover, studies have shown that *Ae. aegypti* is better adapted to urban environments, whereas *Ae. albopictus* prefers peridomestic habitats with denser vegetation [42,43]. *Ae. aegypti* is a highly anthropophilic species and its high abundance observed in the current study may be explained by the massive presence of humans offering large opportunities of blood-feeding for females and high numbers of unmanaged discarded and water storage containers acting as suitable breeding sites for ovipositing, all provided by rapid, uncontrolled and unplanned urbanization [19,21,44].

Our data displayed that *Ae. aegypti*-positive larval breeding sites in the four study areas were highly abundant and diversified, with strong proportions of discarded containers, tires and water storage containers. In Africa, the key container habitats with highest numbers of *Ae. aegypti* pupae and/or larvae are discarded car tires, large domestic water containers (drums and barrels) and small containers (including discarded vessels) in Burkina Faso [45], and jerricans, drums, used or discarded containers and tires in Kenya [46]. In all our study areas, the breeding site positivity was permanently high across the seasons. The most productive larval habitats were water storage containers in SRS, tires in LDS and LDS and water storage containers in SDS. The proportions of *Aedes*-positive breeding sites were directly linked to water storage practices in the domestic areas and rainfalls in the peri-domestic premises, as previously reported in Côte d'Ivoire [19] and Puerto Rico [47]. We observed in the domestic premises that the local populations collected and/or stored water for long duration for various house tasks (e.g., cleaning, cooking, washing, bathing, building and watering plants and animals), to prevent water shortages. People stored potable water for long period, even in the dry season to deal with water interruptions or limited access [47]. This might allow *Ae. aegypti* females to lay their eggs into the water containers that result in the emergence and proliferation of adults [19]. The *Aedes* eggs laid during the dry season could resist to desiccation, and remain viable, and hatch during the next rainy season, thus resulting in an increase in the numbers of larvae and adults [48]. The high *Aedes*-positivity of tires indicates a high proportion of tires infested with *Ae. aegypti* immatures. Indeed, the vehicle tires were used for producing of the local dish "*Attiéké*", roofing, and decoration in domestic areas, and sold or abandoned in the markets and at roadsides in the peridomestic areas [19]. The high abundances of tires and discarded containers might be attributed to the poor management of solid or plastic waste and the lack of community awareness [39,49–51]. Unmanaged tires and discarded cans are more stable as they are less subjected to human disturbance and pressure due to a poor environmental sanitation service. Tires are suitable breeding sites for *Ae. aegypti* larvae due to their ability to hold

water and preserve water for long time and provide shade [39,49,50]. Water in tires is rich in organic detritus and microbial organisms that are an adequate food source for rapid development [50]. Additionally, the temperature, humidity and hiddenness inside tires create a favorable environment for the best development of *Aedes* larvae to pupae, and then the proliferation of adult populations [39]. Overall, the high presence of breeding sites correlated with high abundances of *Ae. aegypti* immature and adults of in the four study sites [52].

Our results demonstrated that all the study sites were exposed to high and similar risks of transmission of DEN and YF viruses, although the epidemics are generally localized and restricted to the health district of Cocody-Bingerville where Anono and Gbagba are located. Moreover, the potential entomological risk indices were above the WHO-established epidemic thresholds in the four study areas. The potential risk of the emergence of a DEN epidemic remained high whatever the seasons and the study sites, while the risk of YF outbreaks was high in the rainy seasons, and medium or low in dry seasons in all the study areas. The high values of the *Aedes* larval and arboviral risk indices suggest that the entire Abidjan city is possibly exposed to large epidemic threats, even if sporadic DEN and YF epidemics have appeared only in some places so far. Indeed, although the risk of an epidemic was medium or low, it was more than sufficient for an epidemic of DEN or YF to occur in all the study areas as an outbreak may occur even if the epidemic risk index is lower than the WHO-established threshold [30]. Indeed, local populations continued to create and maintain *Aedes* breeding sites, despite awareness campaigns as a part of the public health responses. These awareness campaigns were a part of public health responses and were conducted under the aegis of the MHPH through the NIPH [9]. The campaigns consisted of sensitizing, mobilizing and engaging the local populations with the supports of political, religious and community leaders for managing, destroying, removing, or insecticide-treating solid and plastic waste serving as *Aedes* breeding grounds (discarded tires, cans, etc.) and cleaning and covering piped water storage recipients [24,48]. For the 2017-outbreak responses, over 17,000 households were inspected and over 250,000 *Aedes* larval breeding sites were eliminated and/or treated with insecticides [9]. *Stegomyia* indices were very high and statistically comparable between the study areas, independently from the difference in the numbers of reported DEN and YF cases. Therefore, actual epidemic and epidemic-free zones should be included into the arbovirus vector surveillance and control programs.

Our results showed that AHHs of *Ae. aegypti* did not statistically differ between the four study sites. To our experience, AHHs were potentially high (5.72 *Aedes*/house/hour), demonstrating the strong anthropophilic habits of local *Ae. aegypti*. Resting and blood-fed females were mostly collected outdoors in all the study sites, probably due to their diurnal activities, and exophagic and exophilic habits [53–57]. The increased number of unfed females suggests a surge in swarming or reproductive activity of *Ae. aegypti* populations during the rainy season. This highlights a clear correlation between *Ae. aegypti* abundance and rainfall patterns, as reported by several authors [46,58]. Newly emerged females are thus more likely to seek blood meals for their initial egg-laying cycles, which heightens the risk of arbovirus transmission. This pattern aligns with the frequent occurrence of epidemics, typically observed during the rainy season [59]. The high numbers of *Ae. aegypti* adults and parous females could be attributable to the environmental and biological characteristics of urban areas that might be favorable to their survival and longevity [53,54]. Their high presence in and around houses within the domestic premises may be due to their strong anthropophilic behaviors, and this could increase the risk of DEN and YF virus transmission to people [53,54]. The current and previous studies did not analyze *Ae. aegypti* blood-meals and hosts in Abidjan city and Côte d'Ivoire. However, a study reported that the *Ae. aegypti* human-blood index (HBI) was higher than 90% in a similar city, Ouagadougou, Burkina Faso a neighboring country [45]. The close

proximity and short distances between of the larval breeding sites and human residencies may increase human-*Aedes* vector contacts, and human-biting, blood-feeding, resting and egg-laying opportunities [60]. Thus, the almost anthropophily of *Ae. aegypti* found here could increase of risks of DEN and YF virus transmission to humans, mainly outdoors of houses in all the study areas.

Our study suggested that the *Stegomyia* indices were not predictive of current patterns of DEN and YF outbreaks, probably due to some limitations that should be addressed. Indeed, no significant differences were found in *Ae. aegypti* larval indices and adult numbers among study areas, despite differences in the numbers of DEN and YF cases. This suggests that the high abundance of *Ae. aegypti* and high epidemic risk indices alone could not be enough to produce an outbreak of DEN or YF. Similarly, previous studies reported no correlations between entomological risk indices and arboviral epidemics in urbanized cities of Burkina Faso [45] and Kenya [46]. Additional investigations using new methods or new tools are needed to address these limitations for better understanding the differential occurrences of DEN and YF cases in Abidjan. This includes, among others, serological diagnostic tests through rapid diagnostic tests (RDTs) or real-time reverse transcriptase PCR (rRT-PCR) for the detection of DEN and YF within the local population. Indeed, we did not analyze the *Ae. aegypti* bloodmeal sources to determine hosts or reservoirs and vector competence for and infection with DEN and YF viruses due to logistical and funding limitations. Moreover, as the differences in arboviral incidences were captured only among clinical cases from hospitals [61], assessing arboviral infections with arboviruses in the whole populations in the study areas are needed to determine the true prevalences. Indeed, arboviral burden is underestimated as infections are often misdiagnosed as malaria, recorded as non-malarial acute febrile illnesses or unidentified fevers due to a lack of technical capacities [61,62]. Some socio-epidemiological factors such as local community culture, beliefs, knowledge, behaviors, needs and priorities and urban poverty may challenge the diagnostics and vector control efforts. Human movements may compromise the identification of the location of *Aedes* human-biting and arbovirus transmission places, since *Ae. aegypti* is a diurnal vector and people can receive arbovirus-infested bites at their work places or schools outside of their residences. Assessment of the urbanization level, habitation type, land-cover type, housing conditions, vegetation, water supply and/or waste management and their interactions with the ecologies of *Ae. aegypti* and arboviruses is required. *Ae. aegypti* preference for ovipositing in domestic *versus* peridomestic, indoor *versus* outdoor, and water storage *versus* discarded containers is suggestive of behavioral and/or genetic variations in the vector populations [63], thus calling for further investigations.

The uncontrolled galloping urbanization of Abidjan city has resulted in numerous artificial breeding sites conducive to *Ae. aegypti* development and persistence. Our data are important for *Aedes* vector control moving away from reactive entomological control operations to more proactive preventative control. Indeed, there are still no well-structured programs dedicated for routine diagnostics, surveillance and treatment for most arboviruses due to a critical lack of financial, technical and logistical resources. While *Aedes* vector control is crucial to prevent DEN and YF virus transmission, stand-alone government outbreak responses mainly based on outdoor sporadic space-spraying showed short-term and limited effectiveness for controlling *Aedes* vectors and arboviral outbreaks. We identified that key *Ae. aegypti* larval habitats were water storage containers, and unmanaged waste materials such as tires and discarded containers abundantly dispersed in the public and private places. Multisectoral collaborations involving decision-makers, policy-makers, municipal authorities, local health authorities, urban planners, citizen scientists, community health workers and local community leaders and members, and community-based clean-up campaigns focusing on appropriate information, education and empowerment programs are essential for sustainable management and recycling of

identified larval breeding containers [64,65]. Such a holistic, integrated and inclusive vector management practices may be effective for the sustainable controls of *Aedes* vectors and arboviral epidemics in the study areas, and more widely in the city of Abidjan.

## Conclusion

The current study conducted in urban areas with high and low DEN and YF incidences within the city of Abidjan, Côte d'Ivoire showed that all study areas were abundantly infested with *Ae. aegypti* immatures and adults. The key larval breeding sites were water storage receptacles, tires and discarded containers mostly found outside of houses and during the rainy seasons. The *Stegomyia* indices were higher than the WHO-established epidemic thresholds and similar between all the study sites, suggesting that *Ae. aegypti* larval indices were not predictive of DEN and YF outbreaks. These results suggest that local communities were exposed to high biting and resting rates of *Ae. aegypti* and high arbovirus transmission risks outdoors. The outcomes improved our understanding of the distributional patterns of *Aedes* vectors and DEN and YF virus transmission risks in space and time within different eco-epidemiological areas. Overall, the findings offer a baseline for future studies to better understand the relationship between *Ae. aegypti* vectors, the observed risk patterns and DEN and YF incidences for cost-effective prevention of these diseases. In the meantime, a community-based larval source management of identified productive containers might reduce *Ae. aegypti* numbers and risks of transmission of arboviruses in Abidjan, and more widely in other sub-Saharan African cities.

## Supporting information

**S1 Fig. The abundance of *Aedes aegypti* adult populations in the study sites within the city of Abidjan, Côte d'Ivoire.** Error bars show the standard error (SE).
(TIF)

**S2 Fig. Seasonal variations in blood-meal statuses in adult females of *Aedes aegypti* in all the study sites.** SRS: short rainy season, LDS: long dry season, LRS: long rainy season, SDS: short dry season. Error bars show the standard error (SE).
(TIF)

**S3 Fig. Seasonal variations in blood-meal statuses in adult females of *Aedes aegypti* in the study site of Anono, Côte d'Ivoire.** SRS: short rainy season, LDS: long dry season, LRS: long rainy season, SDS: short dry season. Error bars show the standard error (SE).
(TIF)

**S4 Fig. Seasonal variations in blood-meal statuses in adult females of *Aedes aegypti* in the study site of Ayakro, Côte d'Ivoire.** SRS: short rainy season, LDS: long dry season, LRS: long rainy season, SDS: short dry season. Error bars show the standard error (SE).
(TIF)

**S5 Fig. Seasonal variations in blood-meal statuses in adult females of *Aedes aegypti* in the study site of Entente, Côte d'Ivoire.** SRS: short rainy season, LDS: long dry season, LRS: long rainy season, SDS: short dry season. Error bars show the standard error (SE).
(TIF)

**S6 Fig. Seasonal variations in blood-meal statuses in adult females of *Aedes aegypti* in the study site of Gbagba, Côte d'Ivoire.** SRS: short rainy season, LDS: long dry season, LRS: long rainy season, SDS: short dry season. Error bars show the standard error (SE).
(TIF)

**S7 Fig. Seasonal variation in the parity rates of *Aedes aegypti* in the study sites within Abidjan, Côte d'Ivoire.** SRS: short rainy season, LDS: long dry season, LRS: long rainy season, SDS: short dry season. Error bars show the standard error (SE).
(TIF)

**S1 Table. Seasonal variations of the abundance of the larval breeding sites of *Aedes aegypti* in domestic and peridomestic premises in the study sites within the city of Abidjan, Côte d'Ivoire from August 2019 to July 2020.** WSC: Water storage containers, DC: Discarded containers, NBS: Natural breeding sites, N: number of wet containers inspected, n: *Aedes*-positive containers, PW: Percentage of *Aedes*-positive breeding sites among wet containers, PP: Proportion of each *Aedes*-positive breeding site type among the all Aedes-positive breeding site types. PW and PP are expressed as a percentage (%), na: not applicable, SRS: short rainy season, LDS: long dry season, LRS: long rainy season, SDS: short dry season. Others is the category of breeding containers composed of hole of brick, shoes, tarp, flower pot, wooden box, mortar, sheet metal. Natural breeding site is composed of water on land, leaf axils, snail shell, tree hole.
(DOCX)

**S2 Table. Seasonal variations of the abundance of the larval breeding sites of *Aedes aegypti* collected indoors and outdoors of houses in the study sites within the city of Abidjan, Côte d'Ivoire from August 2019 to July 2020.** WSC: Water storage containers, DC: Discarded containers, NBS: Natural breeding sites, N: number of wet containers inspected, n: *Aedes*-positive containers, PW: Percentage of *Aedes*-positive breeding sites among wet containers, PP: Proportion of each *Aedes*-positive breeding site type among the all *Aedes*-positive breeding site types. PW and PP are expressed as a percentage (%), na: not applicable, SRS: short rainy season, LDS: long dry season, LRS: long rainy season, SDS: short dry season. Others is the category of breeding containers composed of hole of brick, shoes, tarp, flower pot, wooden box, mortar, sheet metal. Natural breeding site is composed of water on land, leaf axils, snail shell, tree hole.
(DOCX)

**S3 Table. Seasonal variations of *Aedes aegypti* adults collected outdoors and indoors of houses in the study sites within the city of Abidjan, Côte d'Ivoire from August 2019 to July 2020.** %: Percentage, n: Number of collected adult *Aedes aegypti* mosquitoes.
(DOCX)

**S4 Table. Seasonal variations of the blood-meal status of female populations of *Aedes aegypti* in the study sites within the city of Abidjan, Côte d'Ivoire from August 2019 to July 2020.** %: percentage, n: number of *Aedes aegypti* mosquitoes, SE: standard error, SRS: short rainy season, LDS: long dry season, LRS: long rainy season, SDS: short dry season.
(DOCX)

**S1 Data. Data of Assessing the ecological patterns of Aedes aegypti in areas with high arboviral risks in the large city of Abidjan, Côte d'Ivoire.**
(XLSX)

## Acknowledgments

The authors would like to thank the administrative authorities for authorizing research in the various communities, the community leaders, the customary authorities and the residents of Anono, Gbagba, Ayakro and Entente.

## Author Contributions

**Conceptualization:** Claver N. Adjobi, Julien Z. B. Zahouli, Maurice A. Adja.

**Data curation:** Claver N. Adjobi, Julien Z. B. Zahouli, Allassane F. Ouattara.

**Formal analysis:** Claver N. Adjobi, Allassane F. Ouattara.

**Funding acquisition:** Julien Z. B. Zahouli.

**Investigation:** Claver N. Adjobi, Julien Z. B. Zahouli.

**Methodology:** Claver N. Adjobi, Julien Z. B. Zahouli, Négnorogo Guindo-Coulibaly, Maurice A. Adja.

**Project administration:** Claver N. Adjobi, Julien Z. B. Zahouli.

**Resources:** Julien Z. B. Zahouli.

**Software:** Claver N. Adjobi, Allassane F. Ouattara.

**Supervision:** Julien Z. B. Zahouli, Maurice A. Adja.

**Validation:** Claver N. Adjobi, Julien Z. B. Zahouli, Maurice A. Adja.

**Visualization:** Claver N. Adjobi, Julien Z. B. Zahouli, Maurice A. Adja.

**Writing – original draft:** Claver N. Adjobi.

**Writing – review & editing:** Claver N. Adjobi, Julien Z. B. Zahouli, Négnorogo Guindo-Coulibaly, Laura Vavassori, Maurice A. Adja.

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
