## [Decision Letter · Decision Letter 0]

17 Apr 2024

Dear Mr Adjobi,

Thank you very much for submitting your manuscript "Assessing the ecological patterns of Aedes aegypti in areas with high arboviral risks in the large city of Abidjan, Côte d’Ivoire" for consideration at PLOS Neglected Tropical Diseases. As with all papers reviewed by the journal, your manuscript was reviewed by members of the editorial board and by several independent reviewers. In light of the reviews (below this email), we would like to invite the resubmission of a significantly-revised version that takes into account the reviewers' comments. 

We cannot make any decision about publication until we have seen the revised manuscript and your response to the reviewers' comments. Your revised manuscript is also likely to be sent to reviewers for further evaluation.

Sincerely,

Prof Olaf Horstick, FFPH(UK)

Academic Editor

Amy Morrison

Section Editor

Reviewer's Responses to Questions

**Key Review Criteria Required for Acceptance?**

**Methods**

-Are the objectives of the study clearly articulated with a clear testable hypothesis stated?

-Is the study design appropriate to address the stated objectives?

-Is the population clearly described and appropriate for the hypothesis being tested?

-Is the sample size sufficient to ensure adequate power to address the hypothesis being tested?

-Were correct statistical analysis used to support conclusions?

-Are there concerns about ethical or regulatory requirements being met?

Reviewer #1: No concerns with Methods

Reviewer #2: The paper's methodology looks good overall. The study's goals are clear, and there's a testable hypothesis. The design of the study seems right for what they're trying to find out, and they've described the group they're studying well. The statistics they've used make sense. There don't seem to be any ethical or regulatory issues either. So, the methodology seems solid.

Reviewer #3: (No Response)

**Results**

-Does the analysis presented match the analysis plan?

-Are the results clearly and completely presented?

-Are the figures (Tables, Images) of sufficient quality for clarity?

Reviewer #1: Results are completely presented but could be clearer ( see general comments)

Reviewer #2: In the results section, it seems like they followed the analysis plan. But they could improve the English language for better understanding. The results are clear and complete, though they should work on making the figures (like tables and images) clearer. So, they need to brush up on their English and improve the quality of their figures.

Reviewer #3: (No Response)

**Conclusions**

-Are the conclusions supported by the data presented?

-Are the limitations of analysis clearly described?

-Do the authors discuss how these data can be helpful to advance our understanding of the topic under study?

-Is public health relevance addressed?

Reviewer #1: Conclusions are supported by data and implications are discussed

Reviewer #2: The paper's conclusions seem based on the data, but the English in that part could be better. They did not mention the limitations. They need to better address why this research matters for public health. So, they need to improve the conclusion, mention limitations, and discuss public health relevance.

Reviewer #3: (No Response)

**Editorial and Data Presentation Modifications?**

Reviewer #1: Specific comments

Lines 116, 118 are references available for the lack of success of control programmes

Line 128 little should read few

Line 134-5 it’s good to see a hypothesis being given but, given this hypothesis, what is the basis of the large differences in dengue rates among the different areas in the study between 2017-23?

Line 179-181 What does ‘Aedes mosquito larvae and adults were sampled among 100 and 10 houses per study site and per survey, respectively.’ Mean i.e. were larvae sampled from 100 and adults from 10 houses? Rephrase to clarify.

Line 183-4 investigated for larvae or larvae and adults?

Line 216-8 was sampling only from the indoors of houses or also outdoors?

Line 333 was should read were

Line 339 appears to just repeat line 333 delete one instance to avoid confusion

341-44 I think this is saying that dry seasons positivity was higher than rainy seasons but can you clarify – text is not very clear and I may be misunderstanding

Line 352 was should be were

Line 353-4 should read natural breeding sites

Line 385 Overall, Ae. aegypti positive breeding sites were observed inside of houses, but at a few proportion could read instead Overall, a small proportion of Ae. aegypti positive breeding sites were observed inside of houses

Figure 2 and table 4 appear to contain almost exactly the same information therefore suggest to include only one (I prefer the table) as a main text illustration

Line 490 was should be were

Line 503-7 is this necessary when you have already said there is no significant difference?

Lines 518-520 I don’t understand what is meant here because S1 Fig does not show seasonal variations

Lines 556-565 just seem to repeat the information from Fig 4 and I think could be deleted

Fig4 and 5a-d repeats information in Table 6 and whilst I think Fig 4 is Ok I don’t think non-significant variation warrants a four panel figure (Fig 5) – suggest to delete or move to supplementary

Discussion

Line 650 delete the word especially

Line 652 et should be and

Line 656 delete brackets contents this is repeated a few lines later

Line 663 replace the quasi-only with almost the only

Line 666 an should be a

Line 683 Porto should read Puerto

Line 684 replace with ‘with water for various house cleaning tasks, to prevent water shortages’ 

Line 690 plays should read play

Line 690 second sentence could begin with ‘We observed that …

Line 698 suggest to delete … and therefore less …’ etc.

Line 705 replace , which with that

Line 714 should read either tires or a tire

Line 727 add the before rainy and suggest to delete second sentence

Lines 728-31 need checking an rewriting

Line 741-744 this is a key result for control and needs emphasising e.g. in abstract

Line 752-3 it would be good to give more details about what these awareness campaigns constitute

Line 766 move ‘so-called’ to before epidemic-free (or delete)

Line 771 high relative to what?

Line 777 isn’t the sampling only conducted within houses?

Line 779-780 you did not type the bloodmeals so you are assuming (probably correctly) a high HBI but this assumption needs justifying with respect to literature

Line 785 suggests rather than revealed

Line 814 delete ‘with differential of arboviral indices’

Lines 817-820 need checking and rewriting

Reviewer #2: (No Response)

Reviewer #3: (No Response)

**Summary and General Comments**

Reviewer #1: In this study the ecology of Aedes (primarily aegypti) was assessed in four cites within Abidjan encompassing high and low transmission area across four seasons over one year in 2019-2020. This is a very large dataset and the key result is probably the lack of entomological difference among areas of the city which differ markedly in arbovirus case rates, and limited seasonal variation. This is an important finding for control moving away from reactive entomological control operations to more proactive preventative control when possible. It would be good however for the authors to comment more on any socio-epidemiological factors which might explain the differences in case rates. If entomological indices are not predictive of current patterns are there other factors at play which might be important. This element is absent at present. The Introduction and Methods are well written but the Results are quite densely written with apparently unnecessary repetition of statistics among tables figures and text which makes them quite hard to follow in places. I have made some suggestions below to try to help streamline but I think it would be helpful if the authors checked through and tried to remove unnecessary repetition. Also in several places results are referred to as being higher or lower then followed by a test results which shows no difference; this is better avoided. The Discussion is rather lengthy but does cover all key points however the conclusion needs rewriting.

Reviewer #2: (No Response)

Reviewer #3: An interesting manuscript but which requires some additional clarification and analysis.

Line 31. I’m not sure that it’s outbreak of Aedes mosquito-borne arbovirus. Dengue and yellow fever are diseases or viruses?

Background

Line 82. Aedes-borne viruses? 

Lines 88-89. It would be good to update this information. In 2023146 878 cases of dengue were reported in Burkina-Faso with 688 deaths.

Line 93. arboviral outbreaks????

Lines 93-94. “Aedes aegypti is a key vector of arboviruses in Africa”. Reference is needed.

Line 99-101. Reference

One sentence should be added in the background about dengue control.

Lines 108-109. “Aedes aegypti in Abidjan are resistant to most108 insecticides used for their control [13]”. Have you assessed the insecticide resistance to Aedes in this manuscript?

Methods

Line 148. Reference is needed

The level of urbanization is similar in the four municipalities selected? Type of habitation? Vegetation? Water supplies? Waste management?

How were households selected? There is a difference between households and concession or house?

Line 192. “Aedes mosquito larvae (larvae and pupae)”?????

Culex tigripes???? Please change this by Lutzia tigripes. 

Other Aedes species were found? if yes, please explain the process to establish the proportion of each species.

Line 216. Domestic and peridomestic premises ???? it’is indoor/outdoor?? It is not clear enough.

Line 217. Why was this period selected for adults collection? What was target? Resting mosquitoes?

Line 222. Why 20 individuals?

Line 303. Not clear. Across the manuscript there is sometime confusion between disease and pathogen.

Table 1. it’s possible to determine female and male from larvae

tigripes belong to the genus Lutzia. Please correct it across the manuscript.

What is the potential larval breeding container?

Results

What was the mainly found and most productive larval habitat in rainy season? Or dry season?

What was the proportion of Ae. aegypti collected indoor/outdoor? Why the origin of blood meals ingested by Ae. aegypti was not established? I suggested to the authors to add this important information in the manuscript.

Discussion

Lines 646-648. Reference

Line 660 replace community by population.

Lines 664-666. This is true when Ae. albopictus is absent in the location. Please this sentence needs to be reformulated.

What does it mean strong desiccation?

Please replace entomological risk by potential entomological risk.

Lines 676-680. What is the duration between rainy and dry season?

Lines 683-687. speculation

Please could you discuss the results of the typology of larval habitats considering the observation found elsewhere in Africa and out of Africa?

I would like to draw the authors' attention to the fact that the presence of the vector alone is not enough for there to be an epidemic. It should also be noted that the vectors incriminated here have diurnal activity and people can receive bites outside their place of residence. Please consider this in the discussion.

Line 692-693. “The high positivity of tires is due to the high presence of tires and their use by local communities”. They are using tires to do what?

Lines 803-804. reference

One sentence or paragraph on the absence of Ae. albopictus in Abidjan will be helpful especially since his presence was notified in this city in 2010 (Konan et al. 2013).

PLOS authors have the option to publish the peer review history of their article (what does this mean?). If published, this will include your full peer review and any attached files.

Reviewer #1: No

Reviewer #2: No

Reviewer #3: No
---

## [Decision Letter · Decision Letter 1]

12 Sep 2024

Dear Ph.D Adjobi,

Thank you very much for submitting your manuscript "Assessing the ecological patterns of Aedes aegypti in areas with high arboviral risks in the large city of Abidjan, Côte d’Ivoire" for consideration at PLOS Neglected Tropical Diseases. As with all papers reviewed by the journal, your manuscript was reviewed by members of the editorial board and by several independent reviewers. The reviewers appreciated the attention to an important topic. Based on the reviews, we are likely to accept this manuscript for publication, providing that you modify the manuscript according to the review recommendations. 

Sincerely,

Olaf Horstick, FFPH(UK)

Academic Editor

Amy Morrison

Section Editor

Reviewer's Responses to Questions

**Key Review Criteria Required for Acceptance?**

**Methods**

-Are the objectives of the study clearly articulated with a clear testable hypothesis stated?

-Is the study design appropriate to address the stated objectives?

-Is the population clearly described and appropriate for the hypothesis being tested?

-Is the sample size sufficient to ensure adequate power to address the hypothesis being tested?

-Were correct statistical analysis used to support conclusions?

-Are there concerns about ethical or regulatory requirements being met?

Reviewer #1: Yes to all

Reviewer #3: (No Response)

**Results**

-Does the analysis presented match the analysis plan?

-Are the results clearly and completely presented?

-Are the figures (Tables, Images) of sufficient quality for clarity?

Reviewer #1: Yes to all

Reviewer #2: The results are presented, but additional clarity is needed, particularly in relation to the identification of the sampled mosquitoes. More information on the specific identification techniques and criteria used should be included to support the conclusions. Furthermore, providing the exact WHO references for the risk thresholds would enhance the validity of the results and their interpretation.

Tables need to be more clear. Improving the organization and detail of the tables would strengthen the presentation of the data.

Reviewer #3: (No Response)

**Conclusions**

-Are the conclusions supported by the data presented?

-Are the limitations of analysis clearly described?

-Do the authors discuss how these data can be helpful to advance our understanding of the topic under study?

-Is public health relevance addressed?

Reviewer #1: Yes to all

Reviewer #2: This section also needs improvement. The limitations are not clearly described and no future ideas are presented.

Reviewer #3: (No Response)

**Editorial and Data Presentation Modifications?**

Reviewer #1: (No Response)

Reviewer #2: See attached file

Reviewer #3: (No Response)

**Summary and General Comments**

Reviewer #1: The authors have done a good job with the revision and I just have a few minor comments (below)

Comment on:

Authors’ response: We thank the Reviewer #1 for his question. Please, here we were assessing the entomological risks that are mainly based on the presence and abundance of Aedes vectors. We revised our hypothesis (see revised manuscript, lines 165-167).

Suggest to include the word entomological before risks to clarify the hypothesis and avoid this sounding like a tautology

Comment on:

50) Line 771 high relative to what?

Authors’ response: Please, note that we are not comparing the AHHs to an established index. As AHHs were 5.72 Aedes/house/hour (e.g. with 12.3 Aedes/house/hour in Entente during SRS) (see revised manuscript, Table 5), we consider that AHHs are high based on our experience.

I think the authors need to add this statement or provide some additional support here to make it clear that this is to some extent at least a subjective assessment 

Line numbers refer to manuscript with track changes

Line 164 delete 19 July

Line 502-503 missing ‘of’

Line 779– delete ‘these’

Line 780-782 Given the uninformativeness of the Stegomyia indices and adult abundances (in Discussion and Conclusion), how is the assessment of medium to high levels of risk of transmission assessed? Same comment about lines 905-908 and 913-4. Note that the word potentially is added to line 909 and it would seem appropriate to add this or a similar qualifier elsewhere

Line 842-83. High positivity indicates a high proportion of tires infested rather than a large number of total breeding sites that were tires – may want to rephrase this for clarity

Line 962 the word probably doesn’t fit here

Line 965 delete the word any

Line 970 should ‘T’ be This?

Line 1019 are should be is

Reviewer #2: See attached file

Reviewer #3: Most of my previous comments were addressed but there is still some clarification to be done before accepting the manuscript for publication. 

Line 84. Ae. Aegypti 

Line 91. Aedes aegypti 

Line 94 outbreaks of DEN and YF

Line 121. “This Aedes species can transmit over 50 viruses to humans”. Please double check this.

Lines 175-176. “Abidjan is the first and the ninth largest city of Côte d’Ivoire and Africa, respectively”. Please add the reference.

Lines 345-346. “The tendency of the dominance of Ae. aegypti (99.44%, 83,401/83,870) was observed mainly among the larval collections”. Please could you give the explanation?

Lines 583-587. “In general, the proportions of unfed females were higher during the rainy seasons (57.2 ± 4.1% in LRS and 52.9 ± 4.0% in SRS) compared with the dry seasons (42.0 ± 7.0% in LDS and 37.5 ± 7.0% in SDS) (S2 Fig). In contrast, the proportions of blood-fed females were higher during the dry seasons (30.0 ± 6.5% in LDS and 27.1 ± 6.4% in SDS) than during the rainy seasons (22.8 ± 3.5% in LRS and 17.6 ± 3.1% in SRS)”. This could be due to what? What is the potential implication in term of DEN or YF transmission? This result need to be discussed.

Lines 596-598. “Aedes aegypti parity rate was higher in Ayakro (51.9 � 4.9%, n = 106), followed by Gbagba (50.0 � 4.9, n = 104), Entente (48.6 � 4.9, n = 105), and Anono (48.1 � 5.5%, n = 81)”. Since the difference is not statistically different you cannot say parity was higher in a location.

Line 604. “The parity rate was no significantly different”. Please replace no by not.

Lines 648-650. A tentative explanation of why Ae. albopictus was not found is needed since it was showed that in several countries infested by Ae. albopictus few months later this species become a dominant species replacing the native species Ae. agypti. 

Line 673-674. “The high Aedes-positivity of tires was due to the high presence of tires”. This explanation is too simplistic. Does make sense for me.

Line 701. These awareness campaigns were a part of public health responses coordinated by the NIPH of the MOH. NIPH? MOH?

Line 724. Anthropophilicity?????

Lines 726-727. Where are the epidemiological data for DEN or YF? 

Line 733. Please could you indicate an example of new tools?

Lines 775-776. “The key larval breeding sites were water storage receptacles, tires and discarded containers mostly found outside of houses and during the rainy seasons”. Please could indicate how the key larval breeding sites were determined?

PLOS authors have the option to publish the peer review history of their article (what does this mean?). If published, this will include your full peer review and any attached files.

Reviewer #1: No

Reviewer #2: No

Reviewer #3: No

Figure Files:

Data Requirements:

Reproducibility:

References

---

## [Editor Report · Decision Letter 2]

14 Oct 2024

Dear Ph.D Adjobi,

Thank you very much for submitting your manuscript "Assessing the ecological patterns of Aedes aegypti in areas with high arboviral risks in the large city of Abidjan, Côte d’Ivoire" for consideration at PLOS Neglected Tropical Diseases. As with all papers reviewed by the journal, your manuscript was reviewed by members of the editorial board and by several independent reviewers. The reviewers appreciated the attention to an important topic. Based on the reviews, we are likely to accept this manuscript for publication, providing that you modify the manuscript according to the review recommendations. 

Sincerely,

Olaf Horstick, FFPH(UK)

Academic Editor

Amy Morrison

Section Editor

Figure Files:

Data Requirements:

Reproducibility:

References

---

## [Editor Report · Decision Letter 3]

23 Oct 2024

Dear Ph.D Adjobi,

We are pleased to inform you that your manuscript 'Assessing the ecological patterns of Aedes aegypti in areas with high arboviral risks in the large city of Abidjan, Côte d’Ivoire' has been provisionally accepted for publication in PLOS Neglected Tropical Diseases.

Best regards,

Olaf Horstick, FFPH(UK)

Academic Editor

Amy Morrison

Section Editor

Shaden Kamhawi

co-Editor-in-Chief

Paul Brindley

co-Editor-in-Chief

---

## [Editor Report · Acceptance letter]

11 Nov 2024

Dear Ph.D Adjobi,

We are delighted to inform you that your manuscript, "Assessing the ecological patterns of Aedes aegypti in areas with high arboviral risks in the large city of Abidjan, Côte d’Ivoire," has been formally accepted for publication in PLOS Neglected Tropical Diseases.

Best regards,

Shaden Kamhawi

co-Editor-in-Chief

Paul Brindley

co-Editor-in-Chief
